# Entropy-MCMC: Sampling from Flat Basins with Ease

**Bolian Li, Ruqi Zhang**
Department of Computer Science, Purdue University, USA
`{li4468,ruqiz}@purdue.edu`

## Abstract

Bayesian deep learning counts on the quality of posterior distribution estimation. However, the posterior of deep neural networks is highly multi-modal in nature, with local modes exhibiting varying generalization performance. Given a practical budget, targeting at the original posterior can lead to suboptimal performance, as some samples may become trapped in "bad" modes and suffer from overfitting. Leveraging the observation that "good" modes with low generalization error often reside in flat basins of the energy landscape, we propose to bias sampling on the posterior toward these flat regions. Specifically, we introduce an auxiliary guiding variable, the stationary distribution of which resembles a smoothed posterior free from sharp modes, to lead the MCMC sampler to flat basins. By integrating this guiding variable with the model parameter, we create a simple joint distribution that enables efficient sampling with minimal computational overhead. We prove the convergence of our method and further show that it converges faster than several existing flatness-aware methods in the strongly convex setting. Empirical results demonstrate that our method can successfully sample from flat basins of the posterior, and outperforms all compared baselines on multiple benchmarks including classification, calibration, and out-of-distribution detection.

## 1 Introduction

The effectiveness of Bayesian neural networks relies heavily on the quality of posterior distribution estimation. However, achieving an accurate estimation of the full posterior is extremely difficult due to its high-dimensional and highly multi-modal nature (Zhang et al., 2020b; Izmailov et al., 2021). Moreover, the numerous modes in the energy landscape typically exhibit varying generalization performance. Flat modes often show superior accuracy and robustness, whereas sharp modes tend to have high generalization errors (Hochreiter & Schmidhuber, 1997; Keskar et al., 2017; Bahri et al., 2022). This connection between the geometry of energy landscape and generalization has spurred many works in optimization, ranging from theoretical understanding (Neyshabur et al., 2017; Dinh et al., 2017; Dziugaite & Roy, 2018; Jiang et al., 2019a) to new optimization algorithms (Mobahi, 2016; Izmailov et al., 2018; Chaudhari et al., 2019; Foret et al., 2020).

However, most of the existing Bayesian methods are not aware of the flatness in the energy landscape during posterior inference (Welling & Teh, 2011; Chen et al., 2014; Ma et al., 2015; Zhang et al., 2020b). Their inference strategies are usually energy-oriented and cannot distinguish between flat and sharp modes that have the same energy values. This limitation can significantly undermine their generalization performance, particularly in practical situations where capturing the full posterior is too costly. In light of this, we contend that prioritizing the capture of flat modes is essential when conducting posterior inference for Bayesian neural networks. This is advantageous for improved generalization as justified by previous works (Hochreiter & Schmidhuber, 1997; Keskar et al., 2017; Bahri et al., 2022). It can further be rationalized from a Bayesian marginalization perspective: within the flat basin, each model configuration occupies a substantial volume and contributes significantly to a more precise estimation of the predictive distribution (Bishop, 2006). Moreover, existing flatness-aware methods often rely on a single solution to represent the entire flat basin (Chaudhari et al., 2019; Foret et al., 2020), ignoring the fact that the flat basin contains many high-performing models. Therefore, Bayesian marginalization can potentially offer significant improvements over flatness-aware optimization by sampling from the flat basins (Wilson, 2020; Huang et al., 2020).

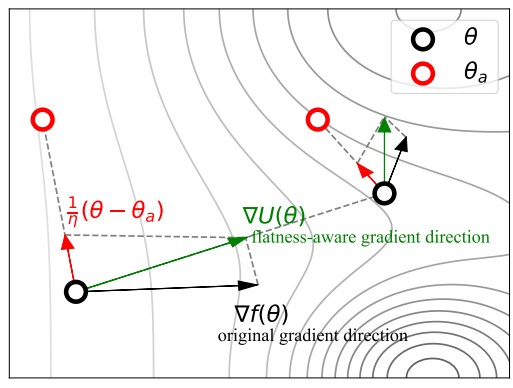 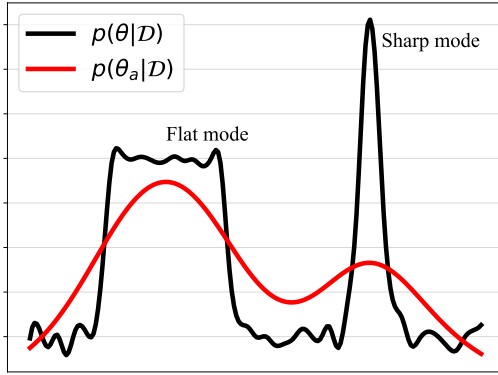

(a) Training dynamics at each step   (b) Original and smoothed posterior distributions

Figure 1: Illustration of Entropy-MCMC. (a) shows how the guiding variable $\boldsymbol{\theta}_a$ pulls $\boldsymbol{\theta}$ toward flat basins; (b) shows two posterior distributions, where $p(\boldsymbol{\theta}_a|\mathcal{D})$ is a smoothed distribution transformed from $p(\boldsymbol{\theta}|\mathcal{D})$, and only keeps flat modes. Entropy-MCMC prioritizes flat modes by leveraging the guiding variable $\boldsymbol{\theta}_a$ from the smoothed posterior as a form of regularization.

Prioritizing flat basins during posterior inference poses an additional challenge to Bayesian inference. Even for single point estimation, explicitly biasing toward the flat basins will introduce substantial computational overhead, inducing nested loops (Chaudhari et al., 2019; Dziugaite & Roy, 2018), doubled gradients calculation (Foret et al., 2020; Möllenhoff & Khan, 2022) or min-max problems (Foret et al., 2020). The efficiency problem needs to be addressed before any flatness-aware Bayesian method becomes practical for deep neural networks.

In this paper, we propose an efficient sampling algorithm to explicitly prioritize flat basins in the energy landscape of deep neural networks. Specifically, we introduce an auxiliary guiding variable $\boldsymbol{\theta}_a$ into the Markov chain to pull model parameters $\boldsymbol{\theta}$ toward flat basins at each updating step (Fig. 1a). $\boldsymbol{\theta}_a$ is sampled from a smoothed posterior distribution which eliminates sharp modes based on local entropy (Baldassi et al., 2016) (Fig. 1b). $\boldsymbol{\theta}_a$ can also be viewed as being achieved by Gaussian convolution, a common technique in diffusion models (Sohl-Dickstein et al., 2015; Song & Ermon, 2019). Our method enjoys a simple joint distribution of $\boldsymbol{\theta}$ and $\boldsymbol{\theta}_a$, and the computational overhead is similar to Stochastic gradient Langevin dynamics (SGLD) (Welling & Teh, 2011). Theoretically, we prove that our method is guaranteed to converge faster than some common flatness-aware methods (Chaudhari et al., 2019; Dziugaite & Roy, 2018) in the strongly convex setting. Empirically, we demonstrate that our method successfully finds flat basins efficiently across multiple tasks. Our main contributions are summarized as follows:

- We propose Entropy-MCMC (EMCMC) for sampling from flat basins in the energy landscape of deep neural networks. EMCMC utilizes an auxiliary guiding variable and a simple joint distribution to efficiently steer the model toward flat basins.

- We prove the convergence of EMCMC and further show that it converges faster than several existing flatness-aware methods in the strongly convex setting.

- We provide extensive experimental results to demonstrate the advantages of EMCMC in sampling from flat basins. EMCMC outperforms all compared baselines on classification, calibration, and out-of-distribution detection with comparable overhead akin to SGLD. We release the code at `https://github.com/lblaoke/EMCMC`.

## 2   RELATED WORKS

**Flatness-aware Optimization.**   The concept of flatness in the energy landscape was first studied by Hochreiter & Schmidhuber (1994), and its connection with generalization was then empirically discussed by Keskar et al. (2017); Dinh et al. (2017); Jiang et al. (2019b). To pursue flatness for better generalization, Baldassi et al. (2015) proposed the local entropy to measure the flatness of local modes, Baldassi et al. (2016) used "replicated" models to implement local entropy, Entropy-SGD (Chaudhari et al., 2019) introduced a nested chain to approximate the local entropy, SAM (Foret et al., 2020) developed a new optimizer to minimize the worst-case near the current model, bSAM (Möllenhoff & Khan, 2022) further improved SAM with a Bayes optimal convex lower bound, LPF (Bisla et al., 2022) introduced low-pass filter to actively search flat basins, and

SWA (Izmailov et al., 2018) found that averaging weights along the trajectory of SGD training can also find flatter modes. Our Entropy-MCMC follows the local entropy measurement and collects more than a single point to fully exploit the flat basins. For detailed comparisons with prior works considering local entropy, please refer to Appendix B.

**MCMC on Deep Neural Networks.** Markov chain Monte Carlo is a class of general and practical sampling algorithms (Andrieu et al., 2003), which has been applied to infer Bayesian neural network posteriors (Neal, 2012). SGMCMC (Welling & Teh, 2011; Ma et al., 2015) methods use the mini-batching technique to adapt MCMC to deep neural networks. SGHMC (Chen et al., 2014) exploited the second-order Langevin dynamics to calibrate the stochastic estimates of HMC gradients. cSGMCMC (Zhang et al., 2020b) further improves sampling efficiency by leveraging a cyclical step size schedule. Symmetric Split HMC (Cobb & Jalaian, 2021) developed a way to apply HMC to deep neural networks without stochastic gradients. Our Entropy-MCMC builds upon the SGMCMC framework and is designed to favor the flat basins in the energy landscape during sampling.

## 3 PRELIMINARIES

**Flatness-aware Optimization.** One common flatness-aware optimization technique is to use the concept of *local entropy*, which measures the geometric properties of the energy landscape (Baldassi et al., 2016; Chaudhari et al., 2019). The local entropy is computed by:

$$\mathcal{F}(\boldsymbol{\theta}; \eta) = \log \int_{\boldsymbol{\Theta}} \exp\left\{-f(\boldsymbol{\theta}') - \frac{1}{2\eta}\|\boldsymbol{\theta} - \boldsymbol{\theta}'\|^2\right\} d\boldsymbol{\theta}', \tag{1}$$

where $f(\cdot)$ is the loss function computed over the entire dataset and $\eta$ is a scalar. The local entropy of a point $\boldsymbol{\theta}$ is determined by its neighbors weighted by their distances, which considers the volume of local modes. Previous optimization methods minimize $-\mathcal{F}(\boldsymbol{\theta}; \eta)$ to find the flat minimum.

**SGMCMC.** Given a dataset $\mathcal{D}$, a neural network with parameters $\boldsymbol{\theta} \in \mathbb{R}^d$, the prior $p(\boldsymbol{\theta})$ and the likelihood $p(\mathcal{D}|\boldsymbol{\theta})$, we can use Markov chain Monte Carlo (MCMC) to sample from the posterior $p(\boldsymbol{\theta}|\mathcal{D}) \propto \exp(-U(\boldsymbol{\theta}))$, where the energy function is $U(\boldsymbol{\theta}) = -\sum_{x \in \mathcal{D}} \log p(x|\boldsymbol{\theta}) - \log p(\boldsymbol{\theta})$. However, the computational cost for MCMC with large-scale data is too high to be practical. SGMCMC tackles this problem by stochastic gradient $\nabla U_{\boldsymbol{\Xi}}$ based on a subset of data $\boldsymbol{\Xi} \subseteq \mathcal{D}$. We use Stochastic Gradient Langevin Dynamics (SGLD) (Welling & Teh, 2011) in the paper as the backbone MCMC algorithm, which has the following updating rule:

$$\boldsymbol{\theta} \leftarrow \boldsymbol{\theta} - \alpha \nabla_{\boldsymbol{\theta}} U_{\boldsymbol{\Xi}}(\boldsymbol{\theta}) + \sqrt{2\alpha} \cdot \boldsymbol{\epsilon}, \tag{2}$$

where $\alpha$ is the step size and $\boldsymbol{\epsilon}$ is standard Gaussian noise. Our method can also be implemented by other SGMCMC methods. During testing, Bayesian marginalization is performed to make predictions based on the sample set collected during sampling $\mathcal{S} = \{\boldsymbol{\theta}_j\}_{j=1}^M$ and the predictive distribution is obtained by $p(y|\boldsymbol{x}, \mathcal{D}) = \int p(y|\boldsymbol{x}, \boldsymbol{\theta}) p(\boldsymbol{\theta}|\mathcal{D}) d\boldsymbol{\theta} \approx \sum_{\boldsymbol{\theta} \in \mathcal{S}} p(y|\boldsymbol{x}, \boldsymbol{\theta})$.

## 4 ENTROPY-MCMC

In this section, we present the Entropy-MCMC algorithm. We introduce the guiding variable $\boldsymbol{\theta}_a$ obtained from the local entropy in Section 4.1 and discuss the sampling strategy in Section 4.2.

### 4.1 FROM LOCAL ENTROPY TO FLAT POSTERIOR

While flat basins in the energy landscape are shown to be of good generalization (Hochreiter & Schmidhuber, 1997; Keskar et al., 2017; Bahri et al., 2022), finding such regions is still challenging due to the highly multi-modal nature of the DNN energy landscape. The updating direction of the model typically needs extra force to keep the sampler away from sharp modes (Chaudhari et al., 2019; Foret et al., 2020). To bias sampling to flat basins, we look into the local entropy (Eq. 1), which can eliminate the sharp modes in the energy landscape (Chaudhari et al., 2019).

We begin by the original posterior distribution $p(\boldsymbol{\theta}|\mathcal{D}) \propto \exp(-f(\boldsymbol{\theta})) = \exp\{\log p(\mathcal{D}|\boldsymbol{\theta}) + \log p(\boldsymbol{\theta})\}$, which contains both sharp and flat modes. By replacing the original loss function with

local entropy, we obtain a smoothed posterior distribution in terms of a new variable $\boldsymbol{\theta}_a$:

$$p(\boldsymbol{\theta}_a|\mathcal{D}) \propto \exp \mathcal{F}(\boldsymbol{\theta}_a; \eta) = \int_{\Theta} \exp \left\{ -f(\boldsymbol{\theta}) - \frac{1}{2\eta}\|\boldsymbol{\theta} - \boldsymbol{\theta}_a\|^2 \right\} d\boldsymbol{\theta}. \tag{3}$$

The effect of local entropy on this new posterior is visualized in Fig. 1b. The new posterior measures both the depth and flatness of the mode in $p(\boldsymbol{\theta}|\mathcal{D})$ by considering surrounding energy values. Thereby, $p(\boldsymbol{\theta}_a|\mathcal{D})$ is expected to primarily capture flat modes in the energy landscape, which can be used as the desired external force to revise the updating directions of the model parameter $\boldsymbol{\theta}$. Moreover, the smoothed posterior $p(\boldsymbol{\theta}|\mathcal{D})$ can be regarded as being obtained through Gaussian convolution, a common approach in diffusion models (Sohl-Dickstein et al., 2015; Song & Ermon, 2019). We also show the effect of hyper-parameter $\eta$ on the flatness of $p(\boldsymbol{\theta}_a|\mathcal{D})$ in Appendix A.4.

However, the complex integral in Eq. 3 requires marginalization on the model parameter $\boldsymbol{\theta}$, which poses a non-trivial challenge. Previous works using local entropy usually adopt an inner Markov chain for approximation (Chaudhari et al., 2019; Dziugaite & Roy, 2018), which sacrifices the accuracy in local entropy computation and induces computationally expensive nested loops in training. We tackle this challenge in a simple yet principled manner, eliminating the need for nested loops or approximation. This is achieved by *coupling* $\boldsymbol{\theta} \sim p(\boldsymbol{\theta}|\mathcal{D})$ and $\boldsymbol{\theta}_a \sim p(\boldsymbol{\theta}_a|\mathcal{D})$ into a joint posterior distribution[1], which enjoys a simple form, as discussed in Lemma 1.

**Lemma 1.** *Assume $\widetilde{\boldsymbol{\theta}} = [\boldsymbol{\theta}^T, \boldsymbol{\theta}_a^T]^T \in \mathbb{R}^{2d}$ and $\widetilde{\boldsymbol{\theta}}$ has the following distribution:*

$$p(\widetilde{\boldsymbol{\theta}}|\mathcal{D}) = p(\boldsymbol{\theta}, \boldsymbol{\theta}_a|\mathcal{D}) \propto \exp \left\{ -f(\boldsymbol{\theta}) - \frac{1}{2\eta}\|\boldsymbol{\theta} - \boldsymbol{\theta}_a\|^2 \right\}. \tag{4}$$

*Then the marginal distributions of $\boldsymbol{\theta}$ and $\boldsymbol{\theta}_a$ are the original posterior $p(\boldsymbol{\theta}|\mathcal{D})$ and $p(\boldsymbol{\theta}_a|\mathcal{D})$ (Eq. 3). Further, the density $p(\widetilde{\boldsymbol{\theta}}|\mathcal{D})$ integrates to a finite quantity and thus it is mathematically well-defined.*

This joint posterior offers three key advantages: i) by coupling $\boldsymbol{\theta}$ and $\boldsymbol{\theta}_a$, we avoid the intricate integral computation, and thus remove the requirement of expensive nested training loops and mitigate the MC approximation error; ii) the joint posterior turns out to be surprisingly simple, making it easy to sample from both empirically and theoretically (details discussed in Sections 4.2 and 5); iii) after coupling, $\boldsymbol{\theta}_a$ provides additional paths for $\boldsymbol{\theta}$ to traverse, making $\boldsymbol{\theta}$ reach flat modes efficiently.

## 4.2 Sampling from Flat Basins

We discuss how to sample from the joint posterior distribution (Eq. 4) in this section. We adopt SGLD (Welling & Teh, 2011), a simple stochastic gradient MCMC algorithm that is suitable for deep neural networks, as the backbone of EMCMC sampling. More advanced MCMC algorithms can also be combined with our method. The energy function of the joint parameter variable $\widetilde{\boldsymbol{\theta}}$ is $U(\widetilde{\boldsymbol{\theta}}) = f(\boldsymbol{\theta}) + \frac{1}{2\eta}\|\boldsymbol{\theta} - \boldsymbol{\theta}_a\|^2$, and thus its gradients is given by:

$$\nabla_{\widetilde{\boldsymbol{\theta}}} U(\widetilde{\boldsymbol{\theta}}) = \left[ \begin{array}{c} \nabla_{\boldsymbol{\theta}} U(\widetilde{\boldsymbol{\theta}}) \\ \nabla_{\boldsymbol{\theta}_a} U(\widetilde{\boldsymbol{\theta}}) \end{array} \right] = \left[ \begin{array}{c} \nabla_{\boldsymbol{\theta}} f(\boldsymbol{\theta}) + \frac{1}{\eta}(\boldsymbol{\theta} - \boldsymbol{\theta}_a) \\ \frac{1}{\eta}(\boldsymbol{\theta}_a - \boldsymbol{\theta}) \end{array} \right]. \tag{5}$$

For the model parameter $\boldsymbol{\theta}$, the original gradient direction $\nabla_{\boldsymbol{\theta}} f(\boldsymbol{\theta})$ is revised by $\frac{1}{\eta}(\boldsymbol{\theta} - \boldsymbol{\theta}_a)$ to get the flatness-aware gradient direction $\nabla_{\boldsymbol{\theta}} U(\widetilde{\boldsymbol{\theta}})$, as visualized in Fig. 1a. Importantly, the practical implementation does not require computing $\nabla_{\boldsymbol{\theta}_a} U(\widetilde{\boldsymbol{\theta}})$ through back-propagation, as we can utilize the analytical expression presented in Eq. 5. Therefore, despite $\widetilde{\boldsymbol{\theta}}$ being in a $2d$ dimension, our cost of gradient computation is essentially the *same* as $d$-dimensional models (e.g., standard SGLD).

With the form of the gradients in Eq. 5, the training procedure of EMCMC is straightforward using the SGLD updating rule in Eq. 2. The details are summarized in Algorithm 1. At testing stage, the collected samples $\mathcal{S}$ are used to approximate the predictive distribution $p(y|\boldsymbol{x}, \mathcal{D}) \approx \sum_{\boldsymbol{\theta}_s \in \mathcal{S}} p(y|\boldsymbol{x}, \boldsymbol{\theta}_s)$. Our choice of sampling from the joint posterior distribution using SGLD, rather than a Gibbs-like approach (Gelfand, 2000), is motivated by SGLD's ability to simultaneously update both $\boldsymbol{\theta}$ and $\boldsymbol{\theta}_a$, which is more efficient than alternative updating (see Appendix A for a detailed

---

[1]Although we refer to $p(\widetilde{\boldsymbol{\theta}}|\mathcal{D})$ as a joint "posterior" to denote its dependency on the dataset, it is obtained through coupling rather than Bayes' rule. Thus, it does not have an explicit prior distribution.

discussion). For the sample set $\mathcal{S}$, we collect both $\boldsymbol{\theta}$ and $\boldsymbol{\theta}_a$ after the burn-in period in order to obtain more high-quality and diverse samples in a finite time budget (see Appendix D.2 for the evidences that $\boldsymbol{\theta}$ and $\boldsymbol{\theta}_a$ find the same mode and Appendix E.3 for performance justification).

In summary, thanks to EMCMC's simple joint distribution, conducting sampling in EMCMC is straightforward, and its computational cost is comparable to that of standard SGLD. Despite its algorithmic simplicity and computational efficiency, EMCMC is guaranteed to bias sampling to flat basins and obtain samples with enhanced generalization and robustness.

---

**Algorithm 1:** Entropy-MCMC

---

**Inputs:** The model parameter $\boldsymbol{\theta} \in \boldsymbol{\Theta}$, guiding variable $\boldsymbol{\theta}_a \in \boldsymbol{\Theta}$, and dataset $\mathcal{D} = \{(\boldsymbol{x}_i, y_i)\}_{i=1}^N$;
**Results:** Collected samples $\mathcal{S} \subset \boldsymbol{\Theta}$;
$\boldsymbol{\theta}_a \leftarrow \boldsymbol{\theta}, \mathcal{S} \leftarrow \emptyset$ ;                                       /* Initialize */
**for** *each iteration* **do**
    $\boldsymbol{\Xi} \leftarrow$ A mini-batch sampled from $\mathcal{D}$;
    $U_{\boldsymbol{\Xi}} \leftarrow -\log p(\boldsymbol{\Xi}|\boldsymbol{\theta}) - \log p(\boldsymbol{\theta}) + \frac{1}{2\eta}\|\boldsymbol{\theta} - \boldsymbol{\theta}_a\|^2$;
    $\boldsymbol{\theta} \leftarrow \boldsymbol{\theta} - \alpha \nabla_{\boldsymbol{\theta}} U_{\boldsymbol{\Xi}} + \sqrt{2\alpha} \cdot \boldsymbol{\epsilon}_1$ ;                              /* $\boldsymbol{\epsilon}_1, \boldsymbol{\epsilon}_2 \sim \mathcal{N}(\boldsymbol{0}, \boldsymbol{I})$ */
    $\boldsymbol{\theta}_a \leftarrow \boldsymbol{\theta}_a - \alpha \nabla_{\boldsymbol{\theta}_a} U_{\boldsymbol{\Xi}} + \sqrt{2\alpha} \cdot \boldsymbol{\epsilon}_2$;

    **if** *after burn-in* **then**
        $\mathcal{S} \leftarrow \mathcal{S} \cup \{\boldsymbol{\theta}, \boldsymbol{\theta}_a\}$ ;                                /* Collect samples */
    **end**
**end**

---

## 5 THEORETICAL ANALYSIS

In this section, we provide a theoretical analysis on the convergence rate of Entropy-MCMC and compare it with previous local-entropy-based methods including Entropy-SGD (Chaudhari et al., 2019) and Entropy-SGLD (Dziugaite & Roy, 2018) (used as a theoretical tool in the literature rather than a practical algorithm). We leverage the 2-Wasserstein distance bounds of SGLD, which assumes the target distribution to be smooth and strongly log-concave (Dalalyan & Karagulyan, 2019). While the target distribution in this case is unimodal, it still reveals the superior convergence rate of EMCMC compared with existing flatness-aware methods. We leave the theoretical analysis on non-log-concave distributions for future work. Specifically, we have the following assumptions for the loss function $f(\cdot)$ and stochastic gradients[2]:

**Assumption 1.** *The loss function $f(\boldsymbol{\theta})$ in the original posterior distribution $\pi = p(\boldsymbol{\theta}|\mathcal{D}) \propto \exp(-f(\boldsymbol{\theta}))$ is $M$-smooth and $m$-strongly convex (i.e., $m\boldsymbol{I} \preceq \nabla^2 f(\boldsymbol{\theta}') \preceq M\boldsymbol{I}$).*

**Assumption 2.** *The variance of stochastic gradients is bounded by $\mathbb{E}[\|\nabla f(\boldsymbol{\theta}) - \nabla f_{\boldsymbol{\Xi}}(\boldsymbol{\theta})\|^2] \leq \sigma^2$ for some constant $\sigma > 0$.*

To establish the convergence analysis for EMCMC, we first observe that the smoothness and convexity of the joint posterior distribution $\pi_{\text{joint}}(\boldsymbol{\theta}, \boldsymbol{\theta}_a) = p(\boldsymbol{\theta}, \boldsymbol{\theta}_a|\mathcal{D})$ in Eq. 4 is the same as the original posterior $p(\boldsymbol{\theta}|\mathcal{D})$, which is formally stated in Lemma 2.

**Lemma 2.** *If Assumption 1 holds and $m \leq 1/\eta \leq M$, then the energy function in the joint posterior distribution $\pi_{joint}(\boldsymbol{\theta}, \boldsymbol{\theta}_a) = p(\boldsymbol{\theta}, \boldsymbol{\theta}_a|\mathcal{D})$ is also $M$-smooth and $m$-strongly convex.*

With the convergence bound of SGLD established by Dalalyan & Karagulyan (2019), we derive the convergence bound for EMCMC in Theorem 1.

**Theorem 1.** *Under Assumptions 1 and 2, let $\mu_0$ be the initial distribution and $\mu_K$ be the distribution obtained by EMCMC after $K$ iterations. If $m \leq 1/\eta \leq M$ and the step size $\alpha \leq 2/(m+M)$, the 2-Wasserstein distance between $\mu_K$ and $\pi_{joint}$ will have the following upper bound:*

$$\mathcal{W}_2(\mu_K, \pi_{joint}) \leq (1 - \alpha m)^K \cdot \mathcal{W}_2(\mu_0, \pi) + 1.65(M/m)(2\alpha d)^{1/2} + \frac{\sigma^2 (2\alpha d)^{1/2}}{1.65 M + \sigma\sqrt{m}}. \quad (6)$$

---

[2]Assumption 1&2 are only for the convergence analysis. Our method and experiments are not restricted to the strong convexity.

Comparing Theorem 1 with the convergence bound of SGLD obtained by Dalalyan & Karagulyan (2019), the only difference is the doubling of the dimension, from $d$ to $2d$. Theorem 1 implies that the convergence rate of EMCMC will have at most a minor slowdown by a constant factor compared to SGLD while ensuring sampling from flat basins.

In contrast, previous local-entropy-based methods often substantially slow down the convergence to bias toward flat basins. For example, consider Entropy-SGD (Chaudhari et al., 2019) which minimizes a flattened loss function $f_{\text{flat}}(\boldsymbol{\theta}) = -\mathcal{F}(\boldsymbol{\theta}; \eta) = -\log \int_{\boldsymbol{\Theta}} \exp\left\{-f(\boldsymbol{\theta}') - \frac{1}{2\eta}\|\boldsymbol{\theta} - \boldsymbol{\theta}'\|^2\right\} d\boldsymbol{\theta}'$. We discuss the convergence bound of Entropy-SGD in Theorem 2, which shows how the presence of the integral (and the nested Markov chain induced by it) slows down the convergence.

**Theorem 2.** *Consider running Entropy-SGD to minimize the flattened loss function $f_{\text{flat}}(\boldsymbol{\theta})$ under Assumptions 1 and 2. Assume the inner Markov chain runs $L$ iterations and the 2-Wasserstein distance between the initial and target distributions is always bounded by $\kappa$. Let $f_{\text{flat}}^*$ represent the global minimum value of $f_{\text{flat}}(\boldsymbol{\theta})$ and $E_t := \mathbb{E}f_{\text{flat}}(\boldsymbol{\theta}_t) - f_{\text{flat}}^*$. If the step size $\alpha \leq 2/(m + M)$, then we have the following upper bound:*

$$E_K \leq \left(1 - \frac{\alpha m}{1 + \eta M}\right)^K \cdot E_0 + \frac{A(1 + \eta M)}{2m}, \tag{7}$$

*where $A^2 = (1 - \alpha m)^L \cdot \kappa + 1.65\left(\frac{M + 1/\eta}{m + 1/\eta}\right)(\alpha d)^{1/2} + \frac{\sigma^2 (\alpha d)^{1/2}}{1.65(M + 1/\eta) + \sigma\sqrt{m + 1/\eta}}$.*

Another example is Entropy-SGLD (Dziugaite & Roy, 2018), a theoretical tool established to analyze Entropy-SGD. Its main distinction with Entropy-SGD is the SGLD updating instead of SGD updating in the outer loop. The convergence bound for Entropy-SGLD is established in Theorem 3.

**Theorem 3.** *Consider running Entropy-SGLD to sample from $\pi_{\text{flat}}(\boldsymbol{\theta}) \propto \exp \mathcal{F}(\boldsymbol{\theta}; \eta)$ under Assumptions 1 and 2. Assume the inner Markov chain runs $L$ iterations and the 2-Wasserstein distance between initial and target distributions is always bounded by $\kappa$. Let $\nu_0$ be the initial distribution and $\nu_K$ be the distribution obtained by Entropy-SGLD after $K$ iterations. If the step size $\alpha \leq 2/(m + M)$, then:*

$$\mathcal{W}_2(\nu_K, \pi_{\text{flat}}) \leq (1 - \alpha m)^K \cdot \mathcal{W}_2(\nu_0, \pi_{\text{flat}}) + 1.65\left(\frac{1 + \eta M}{1 + \eta m}\right)(M/m)(\alpha d)^{1/2} + \frac{A(1 + \eta M)}{m}, \tag{8}$$

*where $A^2 = (1 - \alpha m)^L \cdot \kappa + 1.65\left(\frac{M + 1/\eta}{m + 1/\eta}\right)(\alpha d)^{1/2} + \frac{\sigma^2 (\alpha d)^{1/2}}{1.65(M + 1/\eta) + \sigma\sqrt{m + 1/\eta}}$.*

The complete proof of theorems is in Appendix C. Comparing Theorem 1, 2 and 3, we observe that the convergence rates of Entropy-SGD and Entropy-SGLD algorithms are significantly hindered due to the presence of the nested Markov chains, which induces a large and complicated error term $A$. Since $\sigma$ and $\alpha$ are typically very small, the third term in Theorem 1 will be much smaller than both the third term in Theorem 3 and the second term in Theorem 2.

To summarize, the theoretical analysis provides rigorous guarantees on the convergence of Entropy-MCMC and further demonstrates the superior convergence rate of Entropy-MCMC compared to previous methods in the strongly convex setting.

## 6 EXPERIMENTS

We conduct comprehensive experiments to show the superiority of EMCMC. Section 6.1 and 6.3 demonstrate that EMCMC can successfully sample from flat basins. Section 6.2 verifies the fast convergence of EMCMC. Section 6.4 and 6.5 demonstrate the outstanding performance of EMCMC on multiple benchmarks. Following Zhang et al. (2020b), we adopt a cyclical step size schedule for all sampling methods. For more implementation details, please refer to Appendix E.

### 6.1 SYNTHETIC EXAMPLES

To demonstrate EMCMC's capability to sample from flat basins, we construct a two-mode energy landscape $\frac{1}{2}\mathcal{N}([-2, -1]^T, 0.5\boldsymbol{I}) + \frac{1}{2}\mathcal{N}([2, 1]^T, \boldsymbol{I})$ containing a sharp and a flat mode. To make the

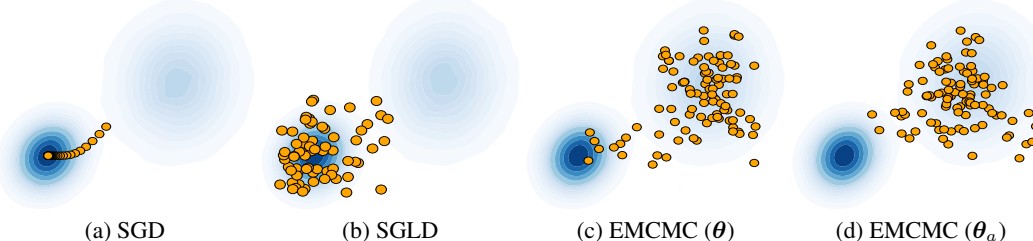

(a) SGD        (b) SGLD        (c) EMCMC ($\boldsymbol{\theta}$)        (d) EMCMC ($\boldsymbol{\theta}_a$)

Figure 2: Sampling trajectories on a synthetic energy landscape with sharp (lower left) and flat (top right) modes. The initial point is located at the ridge of two modes. EMCMC successfully biases toward the flat mode whereas SGD and SGLD are trapped in the sharp mode.

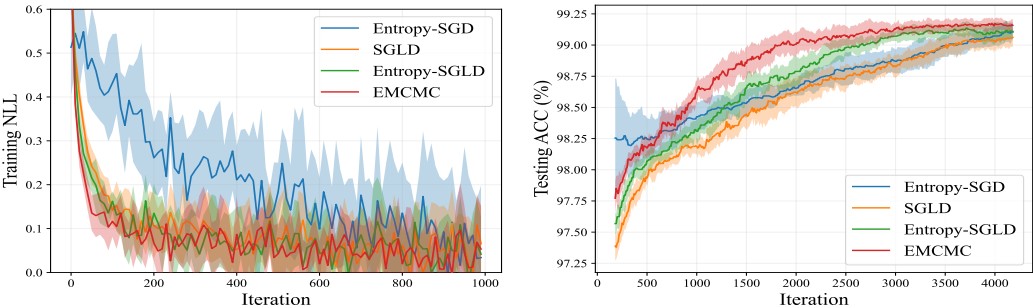

Figure 3: Logistic regression on MNIST in terms of training NLL and testing accuracy (repeated 10 times). EMCMC converges faster than others, which is consistent with our theoretical analysis.

case challenging, we set the initial point at $(-0.2, -0.2)$, the ridge of the two modes[3], which has no strong preference for either mode. The settings for this experiment are: $\eta = 0.5$, $\alpha = 5 \times 10^{-3}$, 1000 iterations, and collecting samples per 10 iterations. Fig. 2 shows that the proposed EMCMC finds the flat basin while SGD and SGLD still prefer the sharp mode due to the slightly larger gradients coming from the sharp mode. From Fig. 2(c)&(d), we see that the samples of $\boldsymbol{\theta}_a$ are always around the flat mode, showing its ability to eliminate the sharp mode. Although $\boldsymbol{\theta}$ visits the sharp mode in the first few iterations, it subsequently inclines toward the flat mode, illustrating the influence of gradient revision by the guiding variable $\boldsymbol{\theta}_a$. If choosing an appropriate $\eta$, EMCMC will find the flat mode no matter how it is initialized. This is due to the stationary distribution of $\boldsymbol{\theta}_a$, which is flattened and removes the sharp mode. Through the interaction term, $\boldsymbol{\theta}_a$ will encourage $\boldsymbol{\theta}$ to the flat mode. We also show the results for different initialization in Appendix D.1.

## 6.2 LOGISTIC REGRESSION

To verify the theoretical results on convergence rates in Section 5, we conduct logistic regression on MNIST (LeCun, 1998) to compare EMCMC with Entropy-SGD Chaudhari et al. (2019), SGLD (Welling & Teh, 2011) and Entropy-SGLD (Dziugaite & Roy, 2018). We follow Maclaurin & Adams (2015) and Zhang et al. (2020a) to use a subset containing 7s and 9s and the resulting posterior is strongly log-concave, satisfying the assumptions in Section 5. Fig. 3 shows that EMCMC converges faster than Entropy-SG(L)D, demonstrating the advantage of using a simple joint distribution without the need for nested loops or MC approximation, which verifies Theorems 1& 2& 3. Besides, while EMCMC and SGLD share similar convergence rates, EMCMC achieves better generalization as shown by its higher test accuracy. This suggests that EMCMC is potentially beneficial in unimodal distributions under limited budgets due to finding samples with high volumes.

## 6.3 FLATNESS ANALYSIS ON DEEP NEURAL NETWORKS

We perform flatness analysis with ResNet18 (He et al., 2016) on CIFAR100 (Krizhevsky, 2009). We use the last sample of SGD, SGLD and EMCMC (averaged result from $\boldsymbol{\theta}$ and $\boldsymbol{\theta}_a$) respectively, and each experiment is repeated 3 times to report the averaged scores.

**Eigenspectrum of Hessian.** The Hessian matrix of the model parameter measures the second-order gradients of a local mode on the energy landscape. Smaller eigenvalues of Hessian indicate a flatter local geometry (Chaudhari et al., 2019; Foret et al., 2020). Since computing the exact Hessian

---

[3]A set of local-maximum points with zero gradients in all directions.

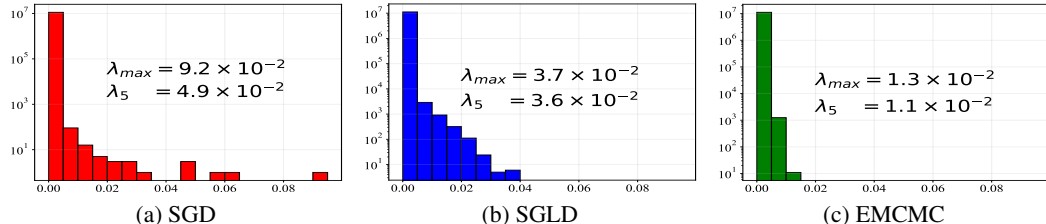

Figure 4: Eigenspectrum of Hessian matrices of ResNet18 on CIFAR100. $x$-axis: eigenvalue, $y$-axis: frequency. A nearly all-zero eigenspectrum indicates a local mode that is flat in all directions. EMCMC successfully finds such flat modes with significantly smaller eigenvalues.

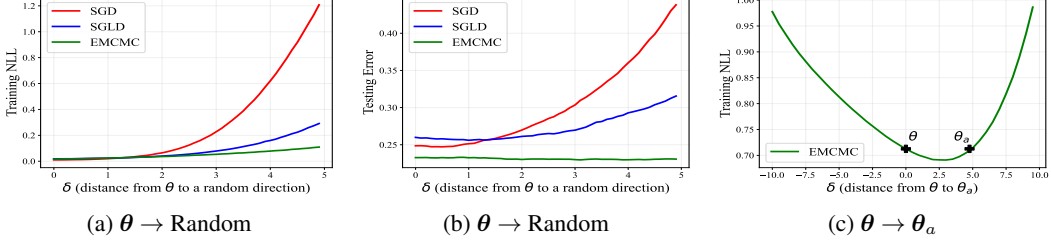

Figure 5: Parameter space interpolation of ResNet18 on CIFAR100. Exploring the neighborhood of local modes from $\boldsymbol{\theta}$ to (a)-(b): a random direction in the parameter space, and (c): $\boldsymbol{\theta}_a$. (a) and (b) show that EMCMC has the lowest and the most flat NLL and error curves. (c) shows that $\boldsymbol{\theta}$ and $\boldsymbol{\theta}_a$ converge to the same flat mode while maintaining diversity.

of deep neural networks is extremely costly due to the dimensionality (Luo et al., 2023), we use the diagonal Fisher information matrix (Wasserman, 2004) to approximate its eigenspectrum:

$$[\lambda_1, ..., \lambda_d]^T \approx diag(\mathcal{I}(\boldsymbol{\theta})) = \mathbb{E}\left[(\nabla U - \mathbb{E}\nabla U)^2\right], \tag{9}$$

where $\lambda_1, ..., \lambda_d$ are eigenvalues of the Hessian. Fig. 4 shows the eigenspectra of local modes discovered by different algorithms. The eigenvalues of EMCMC are much smaller compared with SGD and SGLD, indicating that the local geometry of EMCMC samples is flatter. The eigenspectrum comparison verifies the effectiveness of EMCMC to find and sample from flat basins.

**Parameter Space Interpolation.** Another way to measure the flatness of local modes is directly interpolating their neighborhood in the parameter space (Izmailov et al., 2018). Local modes located in flat basins are expected to have larger widths and better generalization performance (Keskar et al., 2017; Chaudhari et al., 2019). The interpolation begins at $\boldsymbol{\theta}$ and ends at $\boldsymbol{\theta}_\epsilon$ (a random point near $\boldsymbol{\theta}$ or $\boldsymbol{\theta}_\epsilon = \boldsymbol{\theta}_a$). The interpolated point $\boldsymbol{\theta}_\delta$ is computed by:

$$\boldsymbol{\theta}_\delta = (1 - \delta/\|\boldsymbol{\theta} - \boldsymbol{\theta}_\epsilon\|)\,\boldsymbol{\theta} + (\delta/\|\boldsymbol{\theta} - \boldsymbol{\theta}_\epsilon\|)\,\boldsymbol{\theta}_\epsilon, \tag{10}$$

where $\delta$ is the Euclidean distance from $\boldsymbol{\theta}$ to $\boldsymbol{\theta}_\delta$. Fig. 5a and 5b show the training NLL and testing error respectively. The neighborhood of EMCMC maintains consistently lower NLL and errors compared with SGD and SGLD, demonstrating that EMCMC samples are from flatter modes. Furthermore, Fig. 5c visualizes the interpolation between $\boldsymbol{\theta}$ and $\boldsymbol{\theta}_a$, revealing that both variables essentially converge to the same flat mode while maintaining diversity. This justifies the benefit of collecting both of them as samples to obtain a diverse set of high-performing samples.

### 6.4 IMAGE CLASSIFICATION

We conduct classification experiments on CIFAR (Krizhevsky, 2009), corrupted CIFAR (Hendrycks & Dietterich, 2019b) and ImageNet (Deng et al., 2009), to compare EMCMC with both flatness-aware optimization methods (Entropy-SGD (Chaudhari et al., 2019), SAM (Foret et al., 2020) and bSAM (Möllenhoff & Khan, 2022)) and MCMC methods (SGLD (Welling & Teh, 2011) and Entropy-SGLD (Dziugaite & Roy, 2018)). We use ResNet18 and ResNet50 (He et al., 2016) for CIFAR and ImageNet respectively. All sampling algorithms collect a total of 16 samples for Bayesian marginalization, and all entries are repeated 3 times to report the mean±std. Table 1 shows the results on the 3 datasets, in which EMCMC significantly outperforms all baselines. The classification results strongly suggest that by sampling from flat basins, Bayesian neural networks can achieve outstanding performance and EMCMC is an effective and efficient method to do so.

The results for corrupted CIFAR (Hendrycks & Dietterich, 2019a) are shown in Table 1b to show the robustness of EMCMC against multiple types of noises. The results are averaged over all noise types,

Table 1: Classification results on (a) CIFAR10/100, (b) corrupted CIFAR and (c) ImageNet, measured by NLL and accuracy. EMCMC outperforms all compared baselines.

(a) CIFAR10 and CIFAR100

| Method | CIFAR10 | | CIFAR100 | |
|---|---|---|---|---|
| | ACC (%) ↑ | NLL ↓ | ACC (%) ↑ | NLL ↓ |
| SGD | $94.87 \pm 0.04$ | $0.205 \pm 0.015$ | $76.49 \pm 0.27$ | $0.935 \pm 0.021$ |
| Entropy-SGD | $95.11 \pm 0.09$ | $0.184 \pm 0.020$ | $77.45 \pm 0.03$ | $0.895 \pm 0.009$ |
| SAM | $95.25 \pm 0.12$ | $0.166 \pm 0.005$ | $78.41 \pm 0.22$ | $0.876 \pm 0.007$ |
| bSAM | $95.53 \pm 0.09$ | $0.165 \pm 0.002$ | $78.92 \pm 0.25$ | $0.870 \pm 0.005$ |
| SGLD | $95.47 \pm 0.11$ | $0.167 \pm 0.011$ | $78.79 \pm 0.35$ | $0.854 \pm 0.031$ |
| Entropy-SGLD | $94.46 \pm 0.24$ | $0.194 \pm 0.020$ | $77.98 \pm 0.39$ | $0.897 \pm 0.027$ |
| EMCMC | $\mathbf{95.69 \pm 0.06}$ | $\mathbf{0.162 \pm 0.002}$ | $\mathbf{79.16 \pm 0.07}$ | $\mathbf{0.840 \pm 0.004}$ |

(b) Corrupted CIFAR (ACC (%) ↑)

| Severity | 1 | 2 | 3 | 4 | 5 |
|---|---|---|---|---|---|
| SGD | 88.43 | 82.43 | 76.20 | 67.93 | 55.81 |
| SGLD | 88.61 | 82.46 | 76.49 | 69.19 | 56.98 |
| EMCMC | **88.87** | **83.27** | **77.44** | **70.31** | **58.17** |

(c) ImageNet

| Metric | NLL ↓ | Top-1 (%) ↑ | Top-5 (%) ↑ |
|---|---|---|---|
| SGD | 0.960 | 76.046 | 92.776 |
| SGLD | 0.921 | 76.676 | 93.174 |
| EMCMC | **0.895** | **77.096** | **93.424** |

Table 2: OOD detection on CIFAR-SVHN. The predictive uncertainty quantified by EMCMC is the best among the compared algorithms.

| Method | CIFAR10-SVHN | | CIFAR100-SVHN | |
|---|---|---|---|---|
| | AUROC (%) ↑ | AUPR (%) ↑ | AUROC (%) ↑ | AUPR (%) ↑ |
| SGD | **98.30** | **99.24** | 71.96 | 84.08 |
| Entropy-SGD | **98.71** | **99.37** | 79.15 | 86.92 |
| SAM | 94.23 | 95.67 | 74.56 | 84.61 |
| SGLD | 97.66 | 98.64 | 72.51 | 83.35 |
| Entropy-SGLD | 90.07 | 91.80 | 71.83 | 82.89 |
| EMCMC | **98.15** | **99.04** | **81.14** | **87.18** |

and the severity level refers to the strength of noise added to the original data. EMCMC consistently outperforms all compared baselines across all severity levels, indicating that samples from flat basins are more robust to noise. The results for individual noise types are shown in Appendix D.3.

## 6.5 Uncertainty and OOD Detection

To illustrate how predictive uncertainty estimation benefits from flat basins, we evaluate EMCMC on out-of-distribution (OOD) detection. We train each model on CIFAR and quantify uncertainty using the entropy of predictive distributions (Malinin & Gales, 2018). Then we use the uncertainty to detect SVHN samples in a joint testing set combined by CIFAR and SVHN (Netzer et al., 2011). We evaluate each algorithm with Area under ROC Curve (AUROC) (McClish, 1989) and Area under Precision-Recall curve (AUPR) (Olson & Delen, 2008). All other settings remain the same as the classification experiments. Table 2 shows the evaluation results, where EMCMC outperforms nearly all baselines, especially when trained on CIFAR100. This indicates that predictive uncertainty estimation is more accurate if the samples are from flat basins of the posterior. The confidence calibration experiments are shown in Appendix D.4.

## 7 Conclusion and Discussion

We propose a practical MCMC algorithm to sample from flat basins of DNN posterior distributions. Specifically, we introduce a guiding variable based on the local entropy to steer the MCMC sampler toward flat basins. The joint distribution of this variable and the model parameter enjoys a simple form which enables efficient sampling. We prove the fast convergence rate of our method compared with two existing flatness-aware methods. Comprehensive experiments demonstrate the superiority of our method, verifying that it can sample from flat basins and achieve outstanding performance on diverse tasks. Our method is mathematically simple and computationally efficient, allowing for adoption as a drop-in replacement for standard sampling methods such as SGLD.

The results hold promise for both Bayesian methods and deep learning generalization. On the one hand, we demonstrate that explicitly considering flatness in Bayesian deep learning can significantly improve generalization, robustness, and uncertainty estimation, especially under practical computational constraints. On the other hand, we highlight the value of marginalizing over flat basins in the energy landscape, as a means to attain further performance improvements compared to single point optimization methods.

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

# A   ALGORITHM DETAILS

We list some details of the proposed Entropy-MCMC in this section, to help understand our code and reproduction. As discussed in Section 4.2, the updating rule of Entropy-MCMC can be written as:

$$\widetilde{\boldsymbol{\theta}} \leftarrow \widetilde{\boldsymbol{\theta}} - \alpha \nabla_{\widetilde{\boldsymbol{\theta}}} U(\widetilde{\boldsymbol{\theta}}) + \sqrt{2\alpha} \cdot \boldsymbol{\epsilon}, \tag{11}$$

which is a full-batch version. We will show how to apply modern deep learning techniques like mini-batching and temperature to the updating policy in the following sections.

## A.1   MINI-BATCHING

We adopt the standard mini-batching technique in our method, which samples a subset of data points per iteration (Li et al., 2014). We assume the entire dataset to be $\mathcal{D} = \{(\boldsymbol{x}_i, y_i)\}_{i=1}^N$. Then a batch sampled from $\mathcal{D}$ is $\boldsymbol{\Xi} = \{(\boldsymbol{x}_i, y_i)\}_{i=1}^M \subset \mathcal{D}$ with $M \ll N$. For the entire dataset, the loss function is computed by:

$$f(\boldsymbol{\theta}) \propto - \sum_{i=1}^N \log p(y_i | \boldsymbol{x}_i, \boldsymbol{\theta}) - \log p(\boldsymbol{\theta}), \tag{12}$$

and to balance the updating stride per iteration, the loss function for a mini-batch is:

$$f_{\boldsymbol{\Xi}}(\boldsymbol{\theta}) \propto - \frac{N}{M} \sum_{i=1}^M \log p(y_i | \boldsymbol{x}_i, \boldsymbol{\theta}) - \log p(\boldsymbol{\theta}). \tag{13}$$

Therefore, if we average the mini-batch loss over all data points, we can obtain the following form:

$$\bar{f}_{\boldsymbol{\Xi}}(\boldsymbol{\theta}) \propto - \frac{1}{M} \sum_{i=1}^M \log p(y_i | \boldsymbol{x}_i, \boldsymbol{\theta}) - \frac{1}{N} \log p(\boldsymbol{\theta}). \tag{14}$$

If we regard the averaging process as a modification on the stepsize $\alpha$ (i.e., $\bar{\alpha} = \alpha/N$), we will have the following form for the updating policy:

$$\begin{aligned}
\Delta \widetilde{\boldsymbol{\theta}} &= -\bar{\alpha} \cdot \nabla_{\widetilde{\boldsymbol{\theta}}} \widetilde{U}(\widetilde{\boldsymbol{\theta}}) + \sqrt{2\bar{\alpha}} \cdot \boldsymbol{\epsilon} \\
&= -\bar{\alpha} \cdot \nabla_{\widetilde{\boldsymbol{\theta}}} \left[ f_{\boldsymbol{\Xi}}(\boldsymbol{\theta}) + \frac{1}{2\eta} \|\boldsymbol{\theta} - \boldsymbol{\theta}_a\|^2 - \sqrt{\frac{2}{\bar{\alpha}}} \boldsymbol{\epsilon} \odot \widetilde{\boldsymbol{\theta}} \right] \\
&= -\alpha \cdot \nabla_{\widetilde{\boldsymbol{\theta}}} \left[ \bar{f}_{\boldsymbol{\Xi}}(\boldsymbol{\theta}) + \frac{1}{2\eta N} \|\boldsymbol{\theta} - \boldsymbol{\theta}_a\|^2 - \sqrt{\frac{2}{\alpha N}} \boldsymbol{\epsilon} \odot \widetilde{\boldsymbol{\theta}} \right].
\end{aligned} \tag{15}$$

Therefore, the updating rule in Eq.11 can be equivalently written as:

$$\widetilde{\boldsymbol{\theta}} \leftarrow \widetilde{\boldsymbol{\theta}} + \Delta \widetilde{\boldsymbol{\theta}}. \tag{16}$$

## A.2   DATA AUGMENTATION AND TEMPERATURE

We apply data augmentation, which is commonly used in deep neural networks, and compare all methods with data augmentation in the main text. Here, we additionally compare the classification results without data augmentation in Table 3 to demonstrate the effectiveness of EMCMC in this case.

Table 3: Comparison of data augmentation of 3 baselines on CIFAR10. EMCMC outperforms previous methods with and without data augmentation.

| Augmentation | SGD | SGLD | EMCMC |
|:---:|:---:|:---:|:---:|
| × | 89.60 | 89.24 | **89.87** |
| ✓ | 95.59 | 95.64 | **95.79** |

Besides, in the updating policy, a noise term is introduced to add randomness to the sampling process. However, in mini-batch training, the effect of noise will be amplified so that the stationary

distribution of might be far away from the true posterior distribution (Zhang et al., 2020b; Izmailov et al., 2021). Therefore, we also introduce a system temperature $T$ to address this problem.

Formally, the posterior distribution is tempered to be $p(\boldsymbol{\theta}|\mathcal{D}) \propto \exp(-U(\boldsymbol{\theta})/T)$, with an averagely sharpened energy landscape. Similarly, we can regard the temperature effect as a new stepsize $\alpha_T = \alpha/T$, and the updating policy would be:

$$
\begin{aligned}
\Delta\widetilde{\boldsymbol{\theta}} &= -\alpha \cdot \nabla_{\widetilde{\boldsymbol{\theta}}}\left[ \left(\bar{f}_{\boldsymbol{\Xi}}(\boldsymbol{\theta}) + \frac{1}{2\eta N}\|\boldsymbol{\theta} - \boldsymbol{\theta}_a\|^2\right)/T - \sqrt{\frac{2}{\alpha N}}\boldsymbol{\epsilon} \odot \widetilde{\boldsymbol{\theta}} \right] \\
&= -\alpha_T \cdot \nabla_{\widetilde{\boldsymbol{\theta}}}\left[ \bar{f}_{\boldsymbol{\Xi}}(\boldsymbol{\theta}) + \frac{1}{2\eta N}\|\boldsymbol{\theta} - \boldsymbol{\theta}_a\|^2 - \sqrt{\frac{2T}{\alpha_T N}}\boldsymbol{\epsilon} \odot \widetilde{\boldsymbol{\theta}} \right].
\end{aligned}
\tag{17}
$$

To empirically determine how temperature influences classification performances, we compare different temperature levels in Table 4. We compare different temperature magnitudes for both SGLD (Welling & Teh, 2011) and EMCMC. With smaller temperatures, both sampling algorithms improve significantly from $10^0$ to $10^{-4}$, and EMCMC outperforms SGLD under most of temperatures. These findings justify the statement that temperature is needed for good results under data augmentation (Izmailov et al., 2021) and the superiority of EMCMC against other SG-MCMC algorithms.

Table 4: Test accuracy (%) comparison on different temperature magnitudes, with data augmentation, on CIFAR10.

| Temperature | $10^0$ | $10^{-1}$ | $10^{-2}$ | $10^{-3}$ | $10^{-4}$ | $10^{-5}$ | $10^{-6}$ |
|---|---|---|---|---|---|---|---|
| SGLD | 82.89 | 91.67 | 94.12 | 94.95 | 94.86 | 94.88 | 95.17 |
| EMCMC | 81.93 | 91.67 | 94.47 | 94.98 | 95.06 | 95.06 | 95.18 |

### A.3 GIBBS-LIKE UPDATING PROCEDURE

Instead of jointly updating, we can also choose to alternatively update $\boldsymbol{\theta}$ and $\boldsymbol{\theta}_a$. The conditional distribution for the model $\boldsymbol{\theta}$ is:

$$
p(\boldsymbol{\theta}|\boldsymbol{\theta}_a, \mathcal{D}) = \frac{p(\boldsymbol{\theta}, \boldsymbol{\theta}_a|\mathcal{D})}{p(\boldsymbol{\theta}_a|\mathcal{D})} \propto \frac{1}{Z_{\boldsymbol{\theta}_a}} \exp\left\{-f(\boldsymbol{\theta}) - \frac{1}{2\eta}\|\boldsymbol{\theta} - \boldsymbol{\theta}_a\|^2\right\},
\tag{18}
$$

where $Z_{\boldsymbol{\theta}_a} = \exp\mathcal{F}(\boldsymbol{\theta}_a;\eta)$ is a constant. While for the guiding variable $\boldsymbol{\theta}_a$, its conditional distribution is:

$$
p(\boldsymbol{\theta}_a|\boldsymbol{\theta}, \mathcal{D}) = \frac{p(\boldsymbol{\theta}, \boldsymbol{\theta}_a|\mathcal{D})}{p(\boldsymbol{\theta}|\mathcal{D})} \propto \frac{1}{Z_{\boldsymbol{\theta}}} \exp\left\{-\frac{1}{2\eta}\|\boldsymbol{\theta} - \boldsymbol{\theta}_a\|^2\right\},
\tag{19}
$$

where $Z_{\boldsymbol{\theta}} = \exp\left(-f(\boldsymbol{\theta})\right)$ is a constant. Therefore, with Guassian noise, $\boldsymbol{\theta}_a$ is equivalently sampled from $\mathcal{N}(\boldsymbol{\theta}, \eta\boldsymbol{I})$, and the variance $\eta$ controls the expected distance between $\boldsymbol{\theta}$ and $\boldsymbol{\theta}_a$. To obtain samples from the joint distribution, we can sample from $p(\boldsymbol{\theta}|\boldsymbol{\theta}_a, \mathcal{D})$ and $p(\boldsymbol{\theta}_a|\boldsymbol{\theta}, \mathcal{D})$ alternatively. The advantage of doing Gibbs-like updating is that sampling $\boldsymbol{\theta}_a$ can be done exactly. This Gibbs-like updating also shares similarities with the proximal sampling methods (Pereyra, 2016; Lee et al., 2021).

Empirically, we observe that joint updating yields superior performance compared to Gibbs-like updating due to the efficiency of updating both $\boldsymbol{\theta}$ and $\boldsymbol{\theta}_a$ at the same time.

### A.4 EFFECT OF THE VARIANCE TERM ON THE SMOOTHED TARGET DISTRIBUTIONS

We visualize flattened target distributions $p(\boldsymbol{\theta}_a|\mathcal{D})$ under different values of $\eta$ in Fig. 6. As $\eta$ becomes smaller, the target distribution will be flatter.

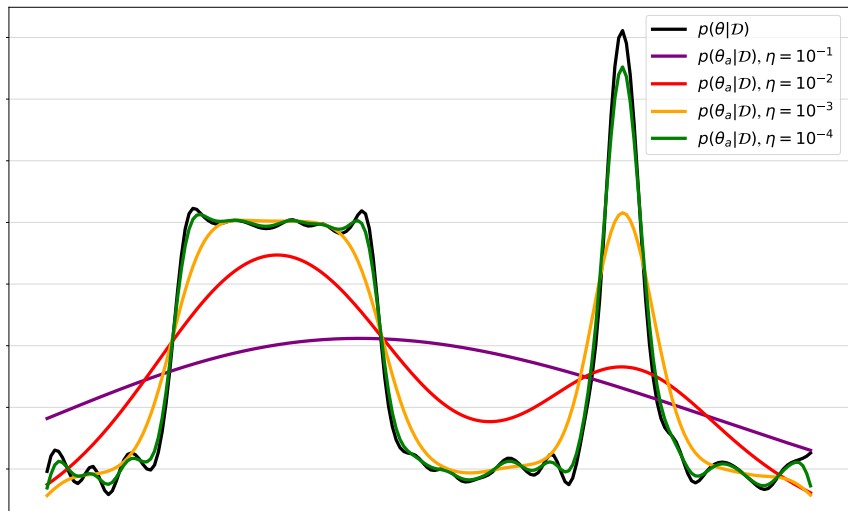

Figure 6: Effect of $\eta$ on the flattened target distribution. Smaller $\eta$ will make the original distribution flatter.

# B  DETAILED COMPARISON WITH RELATED WORKS

## B.1  COMPARISON WITH ROBUST ENSEMBLE

Entropy-MCMC substantially differs from Baldassi et al. (2016) approach and is not a special case of the Robust Ensemble (RE) with $y = 1$ replica. There are several key differences between Entropy-MCMC and RE in Baldassi et al. (2016):

- In Baldassi et al. (2016), the definition of Robust Ensemble, as outlined in Eq. 4, traces out the reference ($\boldsymbol{\theta}_a$ in our paper) and only has $y$ identical replicas in the system, resembling ensemble methods. This can be further verified by Pittorino et al. (2020). As shown in Algorithm 2 in Pittorino et al. (2020), Robust Ensemble with $y = 1$ will essentially be the same as standard SGD. In contrast, Entropy-MCMC has two variables performing different roles, unlike ensemble methods. $\boldsymbol{\theta}_a$ cannot be traced out and plays a crucial role in leading to flat basins during training. Furthermore, Robust Ensemble typically requires at least four models (i.e., $y \geq 3$) while EMCMC only needs two models. Entropy-MCMC greatly reduces computational costs, especially when using large deep neural networks.

- Though Eq. 3 with $y = 1$ in Baldassi et al. (2016) is equivalent to the proposed joint distribution, Baldassi et al. (2016) did not consider the marginal distribution of replicas. In contrast, our work highlights that the marginal of the replica is the original target distribution. This distinction arises because our joint distribution is derived by coupling the original and flattened distributions, a different motivation from Baldassi et al. (2016). Moreover, to the best of our knowledge, Eq. 3 has not been practically applied (as seen in Baldassi et al. (2016) and Pittorino et al. (2020)), primarily serving as an intermediate step in deriving Robust Ensemble (which uses $y$ identical replicas).

- RE is an optimization method that aims to find a flat optimum as the point estimation. In contrast, Entropy-MCMC is a sampling method that aims to sample from the flat basins of the posterior distribution.

In summary, Entropy-MCMC and RE are developed from distinct ideas and they use local entropy in significantly different ways. To the best of our knowledge, the proposed auxiliary guiding variable and the joint distribution's form are novel and non-trivial solutions to flatness-aware learning.

## B.2 COMPARISON WITH ENTROPY-SGD

Entropy-MCMC is not a straightforward change from Chaudhari et al. (2019). It is highly non-trivial to develop an efficient flatness-aware method, as biasing toward the flat basins often introduces substantial computational overhead (Chaudhari et al., 2019; Foret et al., 2020).

A key issue with Chaudhari et al. (2019) approach is its usage of nested Markov chains with Monte Carlo approximation, which significantly reduces convergence speed and estimation accuracy. In contrast, Entropy-MCMC, with an auxiliary guiding variable, guarantees to converge to flat basins with fast speed and minimal overhead.

## C PROOF OF THEOREMS

### C.1 LEMMA 1

*Proof.* Assume $\widetilde{\boldsymbol{\theta}} = [\boldsymbol{\theta}^T, \boldsymbol{\theta}_a^T]^T$ is sampled from the joint posterior distribution:

$$p(\widetilde{\boldsymbol{\theta}}|\mathcal{D}) = p(\boldsymbol{\theta}, \boldsymbol{\theta}_a|\mathcal{D}) \propto \exp\left\{-f(\boldsymbol{\theta}) - \frac{1}{2\eta}\|\boldsymbol{\theta} - \boldsymbol{\theta}_a\|^2\right\}. \tag{20}$$

Then the marginal distribution for $\boldsymbol{\theta}$ is:

$$
\begin{aligned}
p(\boldsymbol{\theta}|\mathcal{D}) &= \int_{\boldsymbol{\Theta}} p(\boldsymbol{\theta}, \boldsymbol{\theta}_a|\mathcal{D}) d\boldsymbol{\theta}_a \\
&= (2\pi\eta)^{-\frac{d}{2}} Z^{-1} \int_{\boldsymbol{\Theta}} \exp\left\{-f(\boldsymbol{\theta}) - \frac{1}{2\eta}\|\boldsymbol{\theta} - \boldsymbol{\theta}_a\|^2\right\} d\boldsymbol{\theta}_a \\
&= Z^{-1} \exp(-f(\boldsymbol{\theta}))(2\pi\eta)^{-\frac{d}{2}} \int_{\boldsymbol{\Theta}} \exp\left\{-\frac{1}{2\eta}\|\boldsymbol{\theta} - \boldsymbol{\theta}_a\|^2\right\} d\boldsymbol{\theta}_a \\
&= Z^{-1} \exp(-f(\boldsymbol{\theta})),
\end{aligned}
\tag{21}
$$

where $Z = \int \exp(-f(\boldsymbol{\theta})) d\boldsymbol{\theta}$ is the normalizing constant, and it is obtained by:

$$\int_{\boldsymbol{\Theta}} \int_{\boldsymbol{\Theta}} \exp\left\{-f(\boldsymbol{\theta}) - \frac{1}{2\eta}\|\boldsymbol{\theta} - \boldsymbol{\theta}_a\|^2\right\} d\boldsymbol{\theta}_a d\boldsymbol{\theta} = (2\pi\eta)^{\frac{d}{2}} \int_{\boldsymbol{\Theta}} \exp(-f(\boldsymbol{\theta})) d\boldsymbol{\theta} := (2\pi\eta)^{\frac{d}{2}} Z. \tag{22}$$

This verifies that the joint posterior distribution $p(\boldsymbol{\theta}, \boldsymbol{\theta}_a|\mathcal{D})$ is mathematically well-defined[4]. Similarly, the marginal distribution for $\boldsymbol{\theta}_a$ is:

$$
\begin{aligned}
p(\boldsymbol{\theta}_a|\mathcal{D}) &= \int_{\boldsymbol{\Theta}} p(\boldsymbol{\theta}, \boldsymbol{\theta}_a|\mathcal{D}) d\boldsymbol{\theta} \\
&\propto \int_{\boldsymbol{\Theta}} \exp\left\{-f(\boldsymbol{\theta}) - \frac{1}{2\eta}\|\boldsymbol{\theta} - \boldsymbol{\theta}_a\|^2\right\} d\boldsymbol{\theta} \\
&= \exp \mathcal{F}(\boldsymbol{\theta}_a; \eta).
\end{aligned}
\tag{23}
$$

$\square$

### C.2 LEMMA 2

*Proof.* Note that we have

$$-\nabla^2 \log \pi_{\text{joint}} = \begin{bmatrix} \nabla^2 f(\theta') + \frac{1}{\eta}I & -\frac{1}{\eta}I \\ -\frac{1}{\eta}I & \frac{1}{\eta}I \end{bmatrix},$$

and after a row reduction, we get

$$\begin{bmatrix} \nabla^2 f(\theta') & 0 \\ -\frac{1}{\eta}I & \frac{1}{\eta}I \end{bmatrix}.$$

---

[4]The exact form of the joint posterior is $p(\boldsymbol{\theta}, \boldsymbol{\theta}_a|\mathcal{D}) = (2\pi\eta)^{-\frac{d}{2}} Z^{-1} \exp(-f(\boldsymbol{\theta}) - \frac{1}{2\eta}\|\boldsymbol{\theta} - \boldsymbol{\theta}_a\|^2)$.

The eigenvalues for this matrix are the eigenvalues of $\nabla^2 f(\theta')$ and $1/\eta$. By the assumption $m \leq 1/\eta \leq M$, we have

$$mI \preceq \begin{bmatrix} \nabla^2 f(\theta') & 0 \\ -\frac{1}{\eta}I & \frac{1}{\eta}I \end{bmatrix} \preceq MI,$$

which means $-\nabla^2 \log \pi_{\text{joint}}$ is also a $M$-smooth and $m$-strongly convex function.

In most scenarios, $m$ often takes on very small values and $M$ tends to be very large. Therefore, the assumption $m \leq 1/\eta \leq M$ is mild. $\qquad\square$

## C.3 Proof of Theorem 1

*Proof.* The proof relies on Theorem 4 from Dalalyan & Karagulyan (2019). Lemma 2 has already provided us with the smoothness and strong convexity parameters for $\pi_{\text{joint}}$. We will now address the bias and variance of stochastic gradient estimation. The stochastic gradient is given by $[\nabla \tilde{f}(\theta') + \frac{1}{\eta}(\theta' - \theta), -\frac{1}{\eta}(\theta' - \theta)]^T$. As $\nabla \tilde{f}(\theta')$ is unbiased and has a variance of $\sigma^2$, the stochastic gradient in our method is also unbiased and has variance $\sigma^2$. Combining the above results, we are ready to apply Theorem 4 (Dalalyan & Karagulyan, 2019) and obtain

$$W_2(\mu_K, \pi_{\text{joint}}) \leq (1 - \alpha m)^K W_2(\mu_0, \pi) + 1.65(M/m)(2\alpha d)^{1/2} + \frac{\sigma^2 (2\alpha d)^{1/2}}{1.65M + \sigma\sqrt{m}}.$$

$\qquad\square$

## C.4 Theorem 3

*Proof.* Let $\pi'(\theta') \propto \exp(-f(\theta') - \frac{1}{2\eta}\|\theta' - \theta\|_2^2)$. It is easy to see that $m + 1/\eta \preceq \nabla^2(-\log \pi') \preceq M + 1/\eta$. Based on Theorem 4 in Dalalyan & Karagulyan (2019), the 2-Wasserstein distance for the inner Markov chain is

$$W_2(\zeta_L, \pi') \leq (1 - \alpha m)^L W_2(\zeta_0, \pi') + 1.65 \left(\frac{M + 1/\eta}{m + 1/\eta}\right)(\alpha d)^{1/2} + \frac{\sigma^2 (\alpha d)^{1/2}}{1.65(M + 1/\eta) + \sigma\sqrt{m + 1/\eta}}$$

$$\leq (1 - \alpha m)^L \kappa + 1.65 \left(\frac{M + 1/\eta}{m + 1/\eta}\right)(\alpha d)^{1/2} + \frac{\sigma^2 (\alpha d)^{1/2}}{1.65(M + 1/\eta) + \sigma\sqrt{m + 1/\eta}}$$

$$:= A^2.$$

Now we consider the convergence of the outer Markov chain. We denote $\pi_{\text{flat}}(\theta) \propto \exp \mathcal{F}(\boldsymbol{\theta}; \eta)$. From Chaudhari et al. (2019), we know that

$$\inf_\theta \left\|\frac{1}{I + \eta\nabla^2 f(\theta)}\right\| mI \preceq -\nabla^2 \log \pi_{\text{flat}} \preceq \sup_\theta \left\|\frac{1}{I + \eta\nabla^2 f(\theta)}\right\| MI.$$

Since $mI \preceq \nabla^2 f(\theta) \preceq MI$, it follows

$$\inf_\theta \left\|\frac{1}{I + \eta\nabla^2 f(\theta)}\right\| \geq \frac{1}{1 + \eta M}, \qquad \sup_\theta \left\|\frac{1}{I + \eta\nabla^2 f(\theta)}\right\| \leq \frac{1}{1 + \eta m}.$$

Therefore,

$$\frac{m}{1 + \eta M}I \preceq -\nabla^2 \log \pi_{\text{flat}} \preceq \frac{M}{1 + \eta m}I.$$

The update rule of the outer SGLD is

$$\theta = \theta - \alpha/\eta(\theta - \mathbf{E}_{\zeta_L}[\theta']) + \sqrt{2\alpha}\xi.$$

The gradient estimation can be written as $\theta - \mathbf{E}_{\pi'}[\theta'] + (\mathbf{E}_{\pi'}[\theta'] - \mathbf{E}_{\zeta_L}[\theta'])$ which can be regarded as the true gradient $\theta - \mathbf{E}_{\pi'}[\theta']$ plus some noise $(\mathbf{E}_{\pi'}[\theta'] - \mathbf{E}_{\zeta_L}[\theta'])$. The bias of the noise can be bounded as follows

$$\|\mathbf{E}_{\pi'}[\theta'] - \mathbf{E}_{\zeta_L}[\theta']\|_2^2 = \left\| \int [\theta'_{\pi'} - \theta'_{\zeta_L}] dJ(\theta'_{\pi'}, \theta'_{\zeta_L}) \right\|_2^2$$

$$\leq \int \left\| \theta'_{\pi'} - \theta'_{\zeta_L} \right\|_2^2 dJ(\theta'_{\pi'}, \theta'_{\zeta_L}).$$

Since the inequality holds for any $J$, we can take the infimum over all possible distributions to conclude

$$\|\mathbf{E}_{\pi'}[\theta'] - \mathbf{E}_{\zeta_L}[\theta']\|_2^2 \leq W_2(\zeta_L, \pi').$$

Furthermore, we note that the variance of the noise is zero. Therefore, by applying Theorem 4 in Dalalyan & Karagulyan (2019) we get

$$W_2(\nu_K, \pi_{\text{flat}}) \leq (1 - \alpha m)^K W_2(\nu_0, \pi_{\text{flat}}) + 1.65 \left( \frac{1 + \eta M}{1 + \eta m} \right) (M/m)(\alpha d)^{1/2} + \frac{A(1 + \eta M)}{m}.$$

$\square$

### C.5 Theorem 2

*Proof.* Compared to Entropy-SGLD, the only difference with Entropy-SGD is doing SGD updates instead of SGLD updates in the outer loop. Therefore, the analysis for the inner Markov chain remains the same as in Theorem 3. To analyze the error of SGD in the outer loop, we follow the results in Ajalloeian & Stich (2020). Since the strongly convex parameter for $f_{\text{flat}}$ is $\frac{m}{1+\eta M}$, by Section 4.2 and Assumption 4 in Ajalloeian & Stich (2020), we know that

$$\frac{1}{2} \|\nabla f_{\text{flat}}(\boldsymbol{\theta}_t)\|^2 \leq \frac{E_t - E_{t+1}}{\alpha} + \frac{1}{2} A$$

$$\Rightarrow \frac{m}{1 + \eta M} E_t \leq \frac{E_t - E_{t+1}}{\alpha} + \frac{1}{2} A$$

$$\Rightarrow E_{t+1} \leq (1 - \frac{\alpha m}{1 + \eta M}) E_t + \frac{1}{2} \alpha A.$$

By unrolling the recursion, we obtain

$$E_K \leq \left( 1 - \frac{\alpha m}{1 + \eta M} \right)^K E_0 + \frac{A(1 + \eta M)}{2m}.$$

$\square$

## D   Additional Experimental Results

We list the additional experimental results in this section, to demonstrate the superiority of our method and show some interesting findings.

### D.1   Additional Synthetic Examples

To demonstrate that EMCMC can bias toward the flat mode under random initialization, we conduct additional synthetic experiments under two different initialization settings. Specifically, we set the initial point to be $(-0.4, -0.4)$ to prefer the sharp mode (Fig. 7) and $(0.0, 0.0)$ to prefer the flat mode (Fig. 8). EMCMC can find the flat mode under all initialization settings, while SGD and SGLD are heavily affected by the choices of initialization.

Besides, we also conduct experiments of running EMCMC for a sufficiently long time to see whether it can converge to the true target distribution (both modes). The results are shown in Fig. 9. $\theta$ successfully finds both modes and $\theta_a$ still samples from the flat mode, both converging to their target distributions. Compared with Fig. 7&8, we are confident to claim that Entropy-MCMC prioritizes flat modes under limited computational budget and will eventually converge to the full posterior with adequate iterations.

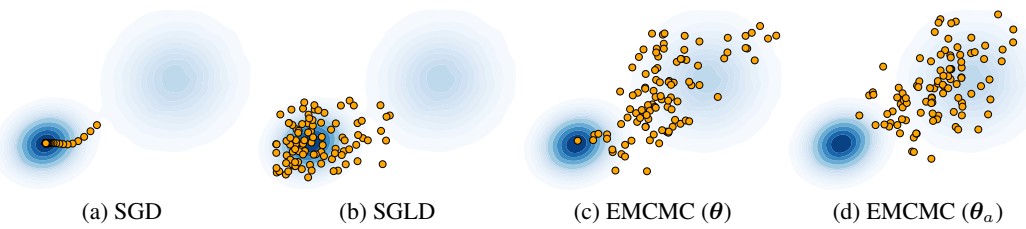

(a) SGD     (b) SGLD     (c) EMCMC ($\boldsymbol{\theta}$)     (d) EMCMC ($\boldsymbol{\theta}_a$)

Figure 7: Synthetic experiments with sharp-mode-biased initialization.

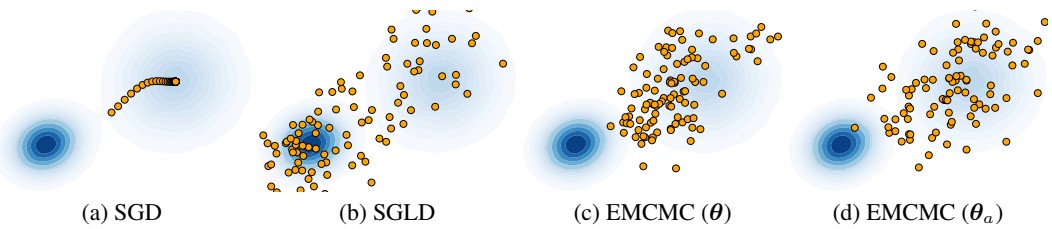

(a) SGD     (b) SGLD     (c) EMCMC ($\boldsymbol{\theta}$)     (d) EMCMC ($\boldsymbol{\theta}_a$)

Figure 8: Synthetic experiments with flat-mode-biased initialization.

## D.2   PARAMETER SPACE INTERPOLATION

As the supplement for Fig. 5, we show additional interpolation results to demonstrate some interesting findings about the model $\boldsymbol{\theta}$ and the auxiliary guiding variable $\boldsymbol{\theta}_a$. The additional interpolation can be separated into the following types:

### D.2.1   TOWARD RANDOM DIRECTIONS

We show the interpolation results toward averaged random directions (10 random directions) in Fig. 10. For the training loss, the auxiliary guiding variable $\boldsymbol{\theta}$ is located at a flatter local region with relatively larger loss values. For the testing error, the guiding variable $\boldsymbol{\theta}_a$ consistently has lower generation errors, which illustrates that the flat modes are preferable in terms of generalization.

### D.2.2   BETWEEN MODEL PARAMETER AND GUIDING VARIABLE

The line between the model parameter $\boldsymbol{\theta}$ and the guiding variable $\boldsymbol{\theta}_a$ is a special direction in the parameter space. The NLL and testing error are both much lower than random directions, which is shown in Fig. 11. Besides, this special direction is biased toward the local region of $\boldsymbol{\theta}_a$, with averagely lower testing errors. This finding justifies the setting of adding $\boldsymbol{\theta}_a$ to the sample set $\mathcal{S}$, since the generalization performance of $\boldsymbol{\theta}_a$ is better.

## D.3   CLASSIFICATION ON CORRUPTED CIFAR

We list the detailed classification results on corrupted CIFAR (Hendrycks & Dietterich, 2018) in Fig. 12, where each corruption type is evaluated at a corresponding subfigure. For the majority of corruption types, our method outperforms other baselines under all severity levels, and is superior especially under severe corruption.

## D.4   CONFIDENCE CALIBRATION

The empirical results for confidence calibration are listed in Table 5. The maximum softmax probabilities are adopted as confidence scores (Hendrycks & Gimpel, 2016), which are then used to predict misclassification over the test set. We use Area under ROC Curve (AUROC) (McClish, 1989) and Expected Calibration Error (ECE) (Naeini et al., 2015) for evaluation. The results indicate that EM-CMC has good calibration capability compared with other baselines, and does not suffer from the overconfidence problem.

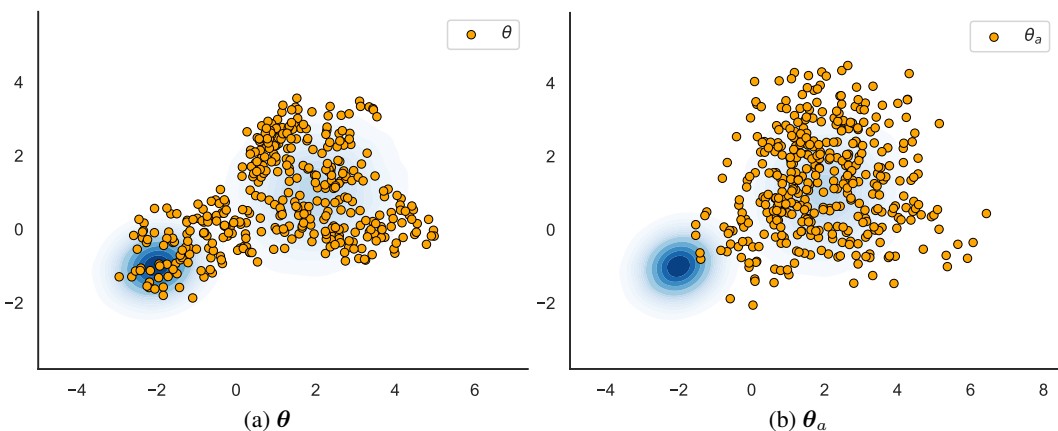

Figure 9: Synthetic experiments with sufficient iterations. Both $\theta$ and $\theta_a$ will fully characterize their target distributions with enough iterations.

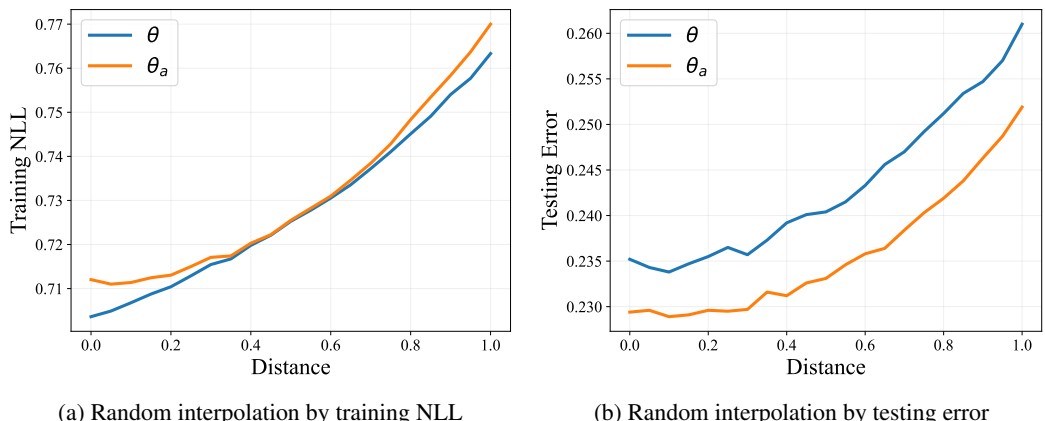

(a) Random interpolation by training NLL    (b) Random interpolation by testing error

Figure 10: Interpolation toward averaged random directions on CIFAR100, comparing the model $\theta$ and the guiding variable $\theta_a$.

# E    ABLATION STUDIES

We empirically discuss several important hyper-parameters and algorithm settings in this section, which justifies our choice of their values. The hyper-parameter choices are tuned via cross-validation, and the ablation studies primarily discuss the influence of these hyper-parameters on empirical results.

## E.1    VARIANCE TERM

We compare different choices of the variance term $\eta$ to determine its influence on the performance. The experimental results are shown in Fig. 13. Generally, setting $\eta$ to be $10^{-3}$ for CIFAR10 and $10^{-2}$ for CIFAR100 will induce outstanding test accuracy. This also implies that the energy landscapes of CIFAR10 and CIFAR100 may be different.

## E.2    STEP SIZE SCHEDULES

We compare different types of stepsize schedules in Table 7. Specifically, we assume the initial and final stepsize to be $\alpha_0$ and $\alpha_1$ respectively. $T$ is the total number of epochs and $t$ is the current

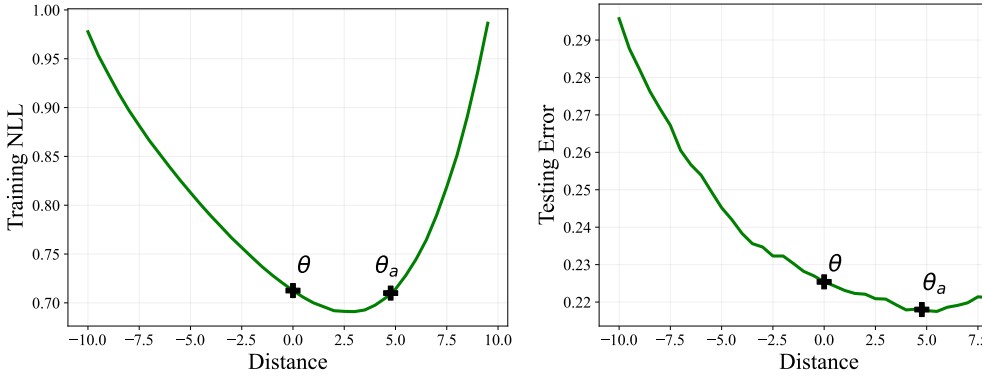

(a) Interpolation between $\boldsymbol{\theta}$ and $\boldsymbol{\theta}_a$ by training NLL     (b) Interpolation between $\boldsymbol{\theta}$ and $\boldsymbol{\theta}_a$ by testing error

Figure 11: Interpolation between the model $\boldsymbol{\theta}$ and the guiding variable $\boldsymbol{\theta}_a$ in terms of training NLL and testing error on CIFAR100.

Table 5: Confidence calibration results on CIFAR. EMCMC has good calibration capability to avoid the overconfidence problem.

| Method | CIFAR10 | | CIFAR100 | |
|---|---|---|---|---|
| | AUROC (%) ↑ | ECE (%) ↓ | AUROC (%) ↑ | ECE (%) ↓ |
| SGD | 92.90 | 2.94 | 87.85 | 6.41 |
| Entropy-SGD | 93.59 | 2.46 | 87.09 | 5.75 |
| SAM | 94.91 | 1.45 | 88.94 | 5.81 |
| SGLD | 94.42 | 0.78 | 87.47 | 2.15 |
| Entropy-SGLD | 93.80 | 0.33 | 87.59 | 1.66 |
| EMCMC | 93.76 | 0.52 | 87.69 | 3.59 |

epoch. The detailed descriptions of stepsize schedules are listed in Table 6. The cyclical stepsize is the best among all stepsize schedules.

Table 6: Formulas or descriptions of different stepsize schedules.

| Name | Formula/Description |
|---|---|
| constant | $\alpha(t) = \alpha_0$ |
| linear | $\alpha(t) = \frac{T-t}{T}(\alpha_0 - \alpha_1) + \alpha_1$ |
| exponential | $\alpha(t) = \alpha_0 \cdot (\alpha_1/\alpha_0)^{t/T}$ |
| step | Remain the same stepsize within one "step", and decay between "steps". |
| cyclical | Follow Eq. 1 in Zhang et al. (2020b). |

### E.3 COLLECTING SAMPLES

Due to the introduction of the auxiliary guiding variable $\boldsymbol{\theta}_a$, the composition of sample set $\mathcal{S}$ has multiple choices: only collect samples of $\boldsymbol{\theta}$, only collect samples of $\boldsymbol{\theta}_a$, collect both samples. We conduct the comparison of all choices and the results are reported in Table 8. It shows that using samples from both $\boldsymbol{\theta}$ and $\boldsymbol{\theta}_a$ gives the best generalization accuracy.

### E.4 NORMALIZATION LAYERS

During testing, the usage of bath normalization layers (BN) in the model architecture induces a problem regarding the mini-batch statistics. The mean and variance of a batch need calculated through at least one forward pass, which is not applicable for the guiding variable $\boldsymbol{\theta}_a$ since it is updated by the distance regularization during training. We try different solutions for this problem,

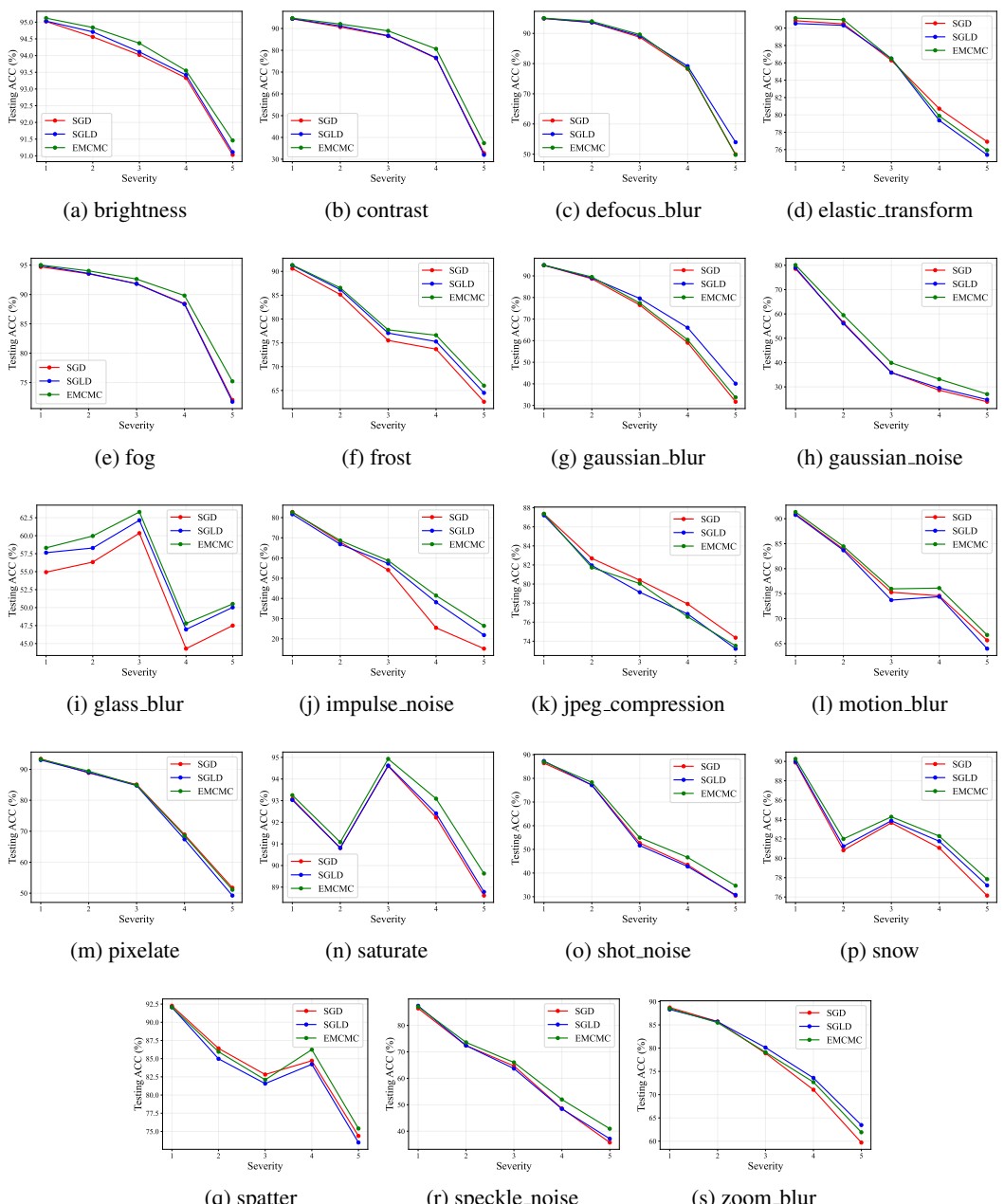

Figure 12: Classification accuracies under different severity levels on corrupted CIFAR. The results are shown per corruption type. Our method outperforms the compared baselines on most of corruption types, especially under high severity levels.

including one additional forward pass and the Filter Response Normalization (Singh & Krishnan, 2020). The comparison is listed in Table 9, where simply adding one additional forward pass during testing can achieve promising accuracy with negligible computational overhead.

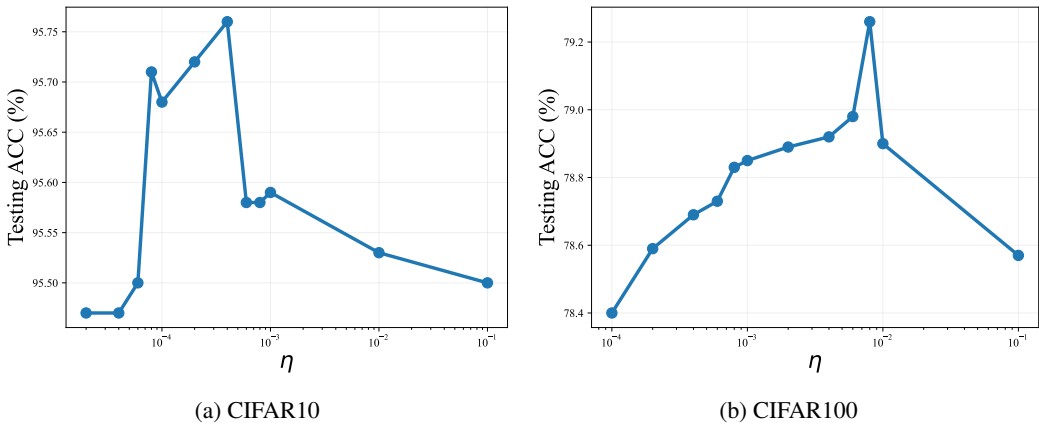

(a) CIFAR10            (b) CIFAR100

Figure 13: Comparison of different variance levels for Entropy algorithm. $\eta \approx 10^{-3}$ is appropriate for CIFAR10, and $\eta \approx 10^{-2}$ is appropriate for CIFAR100.

Table 7: Comparison of stepsize schedules on CIFAR100. The cyclical stepsize is the best for Entropy-MCMC.

| Learning Rate Schedule | constant | linear | exponential | step | cyclical |
|---|---|---|---|---|---|
| Testing ACC (%) ↑ | 88.04 | 87.89 | 87.75 | 89.59 | **89.93** |

### E.5 SGD BURN-IN

We also try SGD burn-in in our ablation studies, by adding the random noise term only to the last few epochs to ensure the fast convergence. We evaluate different settings of SGD burn-in epochs in Table 10. We find that adding 40 burn-in epochs per 50 epochs is the best choice.

Table 8: Ablation study on the composition of sample set $\mathcal{S}$ on CIFAR10. With samples from both $\boldsymbol{\theta}$ and $\boldsymbol{\theta}_a$, the Bayesian marginalization can achieve the best accuracy.

| $\boldsymbol{\theta}$ | $\boldsymbol{\theta}_a$ | ACC (%) ↑ |
|:---:|:---:|:---:|
| ✓ |  | 95.58 |
|  | ✓ | 95.64 |
| ✓ | ✓ | **95.65** |

Table 9: Comparison of different normalization layers on CIFAR10. Simply adding one additional forward pass during testing with standard batch normalization is the best solution.

| Normalization Layer | ACC (%) ↑ | Time (h) |
|:---|:---:|:---:|
| BN | 95.40 | **1.8** |
| BN (one additional forward) | **95.47** | 1.9 |
| FRN (Singh & Krishnan, 2020) | 93.92 | 2.5 |

Table 10: Comparison of different SGD burn-in epochs on CIFAR10. In a 50-epoch round, using SGD burn-in in the first 40 epochs is the best choice.

| SGD Burn-in Epoch | 0 | 10 | 20 | 30 | 40 | 47 |
|:---:|:---:|:---:|:---:|:---:|:---:|:---:|
| Test ACC (%) ↑ | 95.61 | 95.62 | 95.57 | 95.67 | **95.72** | 95.41 |

