# OpenReview forum: "Entropy-MCMC: Sampling from Flat Basins with Ease"
_ICLR.cc/2024/Conference — ICLR 2024 poster_

### Official Review · Reviewer_Prwy · 2023-10-28

**Soundness:** 2 fair
**Presentation:** 2 fair
**Contribution:** 3 good
**Rating:** 5
**Confidence:** 3

**Summary:**

The paper introduces innovative sampling schemes tailored for Bayesian deep learning, aiming to bias the obtained samples during the training process towards flat basins, which helps to obtain better generalization. Rather than focusing on the original posterior, the authors augment it with an auxiliary variable.This augmentation results in a joint distribution with marginals corresponding to the original posterior and a smoothed version of it, achieved through convolution with a Gaussian kernel. The joint distribution's particular structure facilitates straightforward sampling using the stochastic gradient Langevin algorithm. Experimental results demonstrate that acquiring samples from both marginals during the test phase leads to a diverse set of samples that effectively capture the characteristics of flat basins.

**Strengths:**

- The connection between the  proposed posterior augmentation and flatness-aware optimization is quite novel.
- The proposed methodology is very simple, yet effectively improves the performance of BNN.
- The theoretical comparison between EMCMC and entropy-SGD and entropy-SGLD is insightful.
- The reviewer appreciate the extensive set of experiments demonstrating the effectiveness of the method.

**Weaknesses:**

The major weakness of this work is the lack of a rigorous/theoretical justification why the proposed sampling scheme encourages the sampler to explore flat basins. Towards the end of section 4.1, the authors mention that "$\theta_a$ provides additional paths for $\theta$ to traverse, allowing $\theta$ to reach flat modes more efficiently."  However, this argument is not rigorously reasoned in the paper. The only support for this claim seems to be the synthetic examples in section 6.1, which is purely empirical and can have some issues (see Questions section). In my opinion, this claim needs careful theoretical reasoning (even though it can be challenging due to the need to understand the mixing speed of the MCMC chain in non-log-concave settings). This is crucial since it is a key motivation for the methodology. A possible direction to look into is the generalization bound of SGLD, as mentioned in reference [1].

[1] Wang, et.al. 2021, Analyzing the Generalization Capability of SGLD Using Properties of Gaussian Channels

**Questions:**

- In section 6.1, the definition of the ridge between the two modes of the mixture needs clarification. It is also important to explain why choosing an initial point on this ridge does not favor one mode over the other.

- Regarding the MCMC algorithm used in the synthetic example, details about variance parameter $\eta$, number of MCMC iterations, and step sizes are necessary.  Results from Figure 2, 7, and 8 seem to suggest that both SGLD and EMCMC does not mix-well.

- And it would be better to elaborate why the result is independent to the choice of $\eta$? (To my understanding, if $\eta \to 0$, the smoothed target will be identical to the exact target, so no effect on "biasing towards the flat region", and with excessively large $\eta$, the convergence of the MCMC will be very slow.) It will also be helpful to visualize the slice of the target density and smoothed target density (with various choice of $\eta$) on the slice of (x, x).
- This could be due to my limited understanding to Bayesian NN: For the image classification experiement, how do you compare the results obtained from sampling-based and optimization-based algorithm? Specifically, do you obtain certain point estimates of NN's weights from MCMC samples, and compute metrics on test dataset using NN with the estimated weights?
- Is there a systematic way of choosing the variance parameter $\eta$?

---

> ### Author Response · Authors · 2023-11-19
>
> We appreciate your supportive and constructive review. Please find our responses to the questions below.
>
> Q1: The lack of a rigorous/theoretical justification why the proposed sampling scheme encourages the sampler to explore flat basins. In my opinion, this claim needs careful theoretical reasoning. This is crucial since it is a key motivation for the methodology.
>
> A1: We want to clarify that the proposed sampling method does have theoretical justification for the convergence to flat basins. As shown in Lemma 1, the stationary distribution of $\theta_a$ in EMCMC is guaranteed to be the flat target distribution. Consequently, the samples of $\theta_a$ are guaranteed to be from flat basins asymptotically.
>
> Although there is no theoretical guarantee that $\theta$ will visit flat basins before sharp modes, $\theta$ is guaranteed to eventually visit flat basins since its stationary distribution is the original target distribution.
>
> It is important to clarify that we did not claim that $\theta$ is guaranteed to visit flat basins before sharp mode. We claim that $\theta_a$ will _encourage_  $\theta$ to visit flat basins through the interaction term $\frac{1}{2\eta}\\|\theta-\theta_a\\|^2$. This is also verified empirically. See A2 for details.
>
> As acknowledged by the reviewer, proving non-asymptotic behaviors of MCMC methods can be challenging. In fact, many existing papers on improving the non-asymptotic convergence of MCMC do not have rigorous guarantees. For example, papers on improving sampling methods on multi-modal distributions usually only have an asymptotic guarantee and do not have the guarantee to explore multiple modes in a finite time (non-asymptotic convergence), e.g. [4,6,7,8].
>
> While we agree that theoretically proving non-asymptotic behaviors of our method, i.e. $\theta$ converges to flat basins before sharp modes, is a very interesting question, we emphasize that it is not crucial for the proposed methodology. We have provided theoretical guarantees on the convergence to flat basins (Lemma 1) and its fast convergence rate (Theorems 1,2&3). Moreover, besides theoretical results, we provided extensive experimental results to demonstrate the superiority of our method, verifying that it can sample from flat basins and achieve outstanding performance on diverse tasks.
>
> We will clarify this point further in the final version. We hope the significance, novelty, and potential impact of this work can be considered by the reviewer in the final assessment.
>
> Q2: The only support for this claim seems to be the synthetic examples in section 6.1
>
> A2: Besides the results in section 6.1, Fig. 5(c) also demonstrates that $\theta$ and $\theta_a$ both converge to flat modes under practical budgets. In particular, at the same iteration, $\theta$ and $\theta_a$ stay in the same flat mode while maintaining diversity.
>
> Q3: In Section 6.1, the definition of the ridge between the two modes of the mixture needs clarification.
>
> A3: The definition “ridge” in our context is a set of local-maximum points with 0 gradients in all directions. During the initial iterations, the updating is roughly equivalent to adding random noise, with no bias to either mode.
> Therefore, we can compare different methods on their ability to bias towards flat mode. We have also conducted experiments with different initial points (sharp-mode-biased and flat-mode-biased initializations) in Appendix D.1. In all cases, our method can bias to flat basins. We will add this clarification in the final version.
>
> Q4: Regarding the MCMC algorithm used in the synthetic example, details about variance parameter $\eta$, number of MCMC iterations, and step sizes are necessary.
>
> A4: Thanks for pointing this out! We set $\eta=0.5$, the number of iterations to be $1000$, and the step size to be $5e-3$. We collect samples every $10$ iterations. We will add these details in the final version.
>
> Q5: Results from Figure 2, 7, and 8 seem to suggest that both SGLD and EMCMC does not mix well.
>
> A5: In Figures 2, 7, and 8, we aim to test the convergence of these methods under limited budgets, rather than their stationary distributions. Because we know that, given sufficient budgets, SGLD and $\theta$ in EMCMC will converge to the original target distribution and $\theta_a$ in EMCMC will converge to the flat target distribution. This is guaranteed by the asymptotic convergence of MCMC methods.
>
> This synthetic example simulates the scenario in Bayesian deep learning, where sampling methods will not be able to mix well under practical budgets. Mixing on Bayesian neural network posteriors is extremely difficult due to its high-dimensional and highly multi-modal nature [4,5].

---

> ### Author Response · Authors · 2023-11-19
>
> Q6: It would be better to elaborate why the result is independent from the choice of $\eta$.
>
> A6: We did not claim that the results are independent of $\eta$. What we stated in Section 6.1 is “independent of initialization with appropriate $\eta$”. What we mean is the following: if choosing an appropriate $\eta$, EMCMC will find the flat mode no matter how it is initialized. This is because the stationary distribution of $\theta_a$ is the flattened distribution, which removes the sharp modes. Through the interaction term, $\theta_a$ will encourage $\theta$ to flat basins. We will clarify this in the final version.
>
> Q7: It will also be helpful to visualize the slice of the target density and smoothed target density (with various choices of $\eta$).
>
> A7: Please check the following link for this new visualization. We will add it to the final version.
>
> https://anonymous.4open.science/r/EMCMC-FF35/dis_more_eta.pdf
>
> Q8: For the image classification experiments, how do you compare the results obtained from sampling-based and optimization-based algorithms?
>
> A8: Each MCMC sample $\theta_j$ is a neural network model. If the algorithm collects $M$ samples in total, there will be $M$ neural network weights saved in the memory. In the classification experiment, each model $\theta_j$ will output a probability $p(y|x,\theta_j)$, and the final output of sampling-based methods would be:
> $$
> p(y|x,D)\approx\sum_{j=1}^M p(y|x,\theta_j).
> $$
> This is called Bayesian model averaging, which is widely used in Bayesian deep learning [4,5]. In the classification experiment, accuracy, NLL, and ECE of sampling-based methods are obtained via Bayesian model averaging, following previous works [4,5].
>
> Q9: Is there a systematic way of choosing the variance parameter $\eta$?
>
> A9: $\eta$ is chosen by hyperparameter tuning via cross-validation. We found that the performance is not very sensitive to $\eta$ and we only need to determine the appropriate scale (e.g., considering $\eta \in \\{10^{-1},10^{-2},10^{-3},10^{4}\\}$). For example, the accuracy in the scale of $10^{-4}$ is shown in the following link, where no significant difference is found.
>
> https://anonymous.4open.science/r/EMCMC-FF35/eta_CIFAR10_grid.pdf
>
> We agree that a more systematic way of choosing $\eta$ is an interesting question and leave it for future work.
>
> [1] Entropy-SGD: Biasing Gradient Descent into Wide Valleys. ICLR 2017.
>
> [2] Entropic gradient descent algorithms and wide flat minima. ICLR, 2020.
>
> [3] Unreasonable effectiveness of learning neural networks: From accessible states and robust ensembles to basic algorithmic schemes. Proceedings of the National Academy of Sciences, 2016.
>
> [4] Cyclical Stochastic Gradient MCMC for Bayesian Deep Learning. ICLR 2020.
>
> [5] What Are Bayesian Neural Network Posteriors Really Like? ICML 2021.
>
> [6] A Contour Stochastic Gradient Langevin Dynamics Algorithm for Simulations of Multi-modal Distributions. NeurIPS 2020.
>
> [7] Non-convex Learning via Replica Exchange Stochastic Gradient MCMC. ICML 2020.
>
> [8] Interacting Contour Stochastic Gradient Langevin Dynamics. ICLR 2022.

---

> > ### Comment · Reviewer_Prwy · 2023-11-21
> > **Thanks for your response**
> >
> > Thanks for answering my questions. I respectfully disagree with A1 and A4  provided above.
> >
> > - to A1: I understood that the marginal distribution $\theta_a$ is more smooth due to the Gaussian smoothing kernel introduced. However, my question Q1 is asking the rigorous reasoning of the comments after Lemma1, "after coupling, $\theta_a$ provides additional paths for $\theta$ to traverse, making $\theta$ reach flat modes efficiently." This comments on the sample path of $\theta$, whose marginal distribution is the original posterior. And the synthetic example does corroborate the argument.
> > It's also worth clarifying that it is not about whether $\theta$ visits the flat basin before or after it visits sharp basin, the precise question to ask is whether the MCMC samples are less likely to be confined around the sharp basin due to the interacting term.
> >
> > Although I understand that in practice one can also use samples from $\theta_a$ to perform inference (and the author suggested to do that), it is important to note that samples from $\theta_a$ is biased, meaning that it converges to the smoothed posterior intead of the correct one. In fact, the smoothed posterior can be very different to the exact posterior and the mode of the smoothed one may not be on the flat basin (see for example fig 1 of The computational asymptotics of Gaussian variational inference and the Laplace approximation, Zuheng Xu, Trevor Campbell, 2022).
> >
> > - to A5: "This synthetic example simulates the scenario in Bayesian deep learning, where sampling methods will not be able to mix well under practical budgets". This is indeed the major computation challenge of BNN, but should not be a feature that one should consider in evaluating computational method on BNN. Yes, we know that given infinite computational budget, any valid MCMC method will eventually converge; but the real difference between a good and bad MCMC algorithms are their practical mixing speed and how easily the MCMC sample can get rid of the "metastable state" (which refers to the phenomenon that MCMC chain gets trapped in one mode and fail to visit low-probability region). Therefore, one should at least evaluate samples scatters after the chain stabilize. I personally cannot be convinced by results that produced from a chain that does not necessarily converge.
> >
> > Accordingly, I choose to remain the score.

---

> ### Author Response · Authors · 2023-11-21
> **Clarification for your remaining concerns**
>
> Thank you very much for your reply and further clarification on your questions.
>
> Q: Theoretical guarantee for the sentence after Lemma 1.
>
> A: The sentence after Lemma 1 “after coupling, $\theta_a$ provides additional paths to traverse, making $\theta$ reach flat modes efficiently” provides intuition and explanation. Our experiments in Fig.2, Fig. 5(c), and Fig.7&8 have verified empirically that $\theta$ in EMCMC will not be confined around the sharp basin. Note that we did not claim in the paper that this sentence is theoretically guaranteed to hold.
>
> We believe whether having a rigorous theorem for this sentence should not be the reason for rejection. As a conference paper, this work has already provided novel insights on posterior inference and generalization, a highly practical algorithm, a theoretical convergence rate comparison, and comprehensive experimental results.
>
> As outlined in our A1, many papers introducing new methodologies lack rigorous guarantees on mixing speed or non-asymptotic convergence due to the inherent complexity of such proofs.
>
> We kindly hope you can consider the larger context of this work and the contributions it provided in your final evaluation.
>
>
> Q: Evaluate sample scatters after the chain stabilizes
>
> A: We ran SGLD and EMCMC on the synthetic example for a sufficiently long time till they stabilized. Please find the new figures in the links below:
>
> SGLD: https://anonymous.4open.science/r/EMCMC-FF35/sync_SGLD_converge.pdf
>
> EMCMC $\theta$: https://anonymous.4open.science/r/EMCMC-FF35/sync_sampler_converge.pdf
>
> EMCMC $\theta_a$: https://anonymous.4open.science/r/EMCMC-FF35/sync_anchor_converge.pdf
>
> From the results, we see that: 1) EMCMC can mix well since $\theta$ and $\theta_a$ both converge to their stationary distributions respectively. Similarly, SGLD also mixes well in this case. 2) The mixing speed of EMCMC is similar to SGLD (they both mix after about 50000 iterations), which aligns with our Theorem 1 that EMCMC has a similar convergence rate as SGLD.
>
> We will add the results to the revision. We would appreciate it if you could consider raising the score in light of our new results and response.

---

> > ### Comment · Reviewer_Prwy · 2023-11-21
> >
> > I appreciate the authors' prompt response and clarifications.
> >
> > For those additional figures about SGLD and EMCMC, aren't they neutral (if not negative) results? As the original SGLD successfully send particles towards the flat component, and EMCMC shows a similar mixing speed, which do not show any empirical advantages of EMCMC over SGLD.
> >
> > In terms of the theoretical justification, I value rigorous and scientific reasoning on the usefulness of a methodology that has decent performance. I think this is the key to help the community to develep more insights into the general problem.

---

> > > ### Author Response · Authors · 2023-11-21
> > >
> > > We appreciate your prompt reply! Your questions and inquiries have significantly contributed to a clearer understanding of this work.
> > >
> > > As requested, the new figures showed the results of EMCMC and SGLD _after sufficient running time_ until they stabilized. Given sufficient time, they both mix well on the synthetic example. The advantages of EMCMC over SGLD have been clearly demonstrated in our experiments in the paper. Under _practical budgets_, EMCMC can sample from flat basins whereas SGLD cannot.
> > >
> > > We also value methods that both have rigorous guarantees and decent practical performance. Yet, the evolution of algorithms and theory may not always happen at the same time. This is especially true for the field of (Bayesian) deep learning. We believe having an open mind that appreciates various contributions is crucial for the development of the ML community.

---

> ### Author Response · Authors · 2023-11-22
>
> We are grateful for your helpful comments and constructive feedback on our paper. Please kindly let us know if you have any remaining questions or concerns so that we can address them before the deadline.
>
> If you feel that your original concerns have been addressed, we would appreciate it if you could consider raising the score to reflect this. Thank you!

---

> > ### Comment · Reviewer_Prwy · 2023-11-22
> >
> > I appreciate the ongoing discussion. However, I respectfully maintain my original stance and have decided to retain the current score. Regardless, I wish you the best of luck with your submission.

---

> > > ### Author Response · Authors · 2023-11-22
> > >
> > > Thank you again for your engagement in the discussion!

---

### Official Review · Reviewer_JBLt · 2023-10-30

**Soundness:** 3 good
**Presentation:** 3 good
**Contribution:** 3 good
**Rating:** 6
**Confidence:** 3

**Summary:**

This paper proposes entropy-MCMC (EMCMC), a method to bias posterior sampling from Bayesian neural networks towards flat basins to improve generalization performance. EMCMC works by introducing an auxiliary guiding variable with a smoothed posterior that favors flat basins. The paper presents some theoretical analysis on convergence rates, and demonstrates the effectiveness with experiments on multiple benchmarks.

**Strengths:**

* The paper tackles an important problem, naming find modes in flat basins to improve generalization.
* The paper is clearly written, and easy to follow.
* The core idea of the paper is quite simple: instead of relying on an inner loop to estimate the gradients of the local entropy as done in Entropy-SGD, this paper proposes to remove the integral and introduce an auxiliary variable, and sample from the new joint distribution which has a simple form and can be done in a computationally efficient way. Despite its simplicity, the idea is interesting and quite natural. Theoretical analysis and empirical results both demonstrate the effectiveness of the proposed approach.
* The theoretical results look interesting but I did not carefully check the math.

**Weaknesses:**

This paper presents the method as a sampling/MCMC approach, which seems a bit confusing to me. When we talk about sampling we usually aim to sample from the true underlying distribution, including both the flat basins and the sharp nodes. However, it seems in the context of this paper, it is perfectly fine to be stuck at a good flat basin without ever exploring around the sharp nodes. Moreover, Theorem 1 presents a convergence result, implying that the marginal distribution of \theta should eventually converge to the true posterior distribution, but this means it would have to eventually visit the sharp modes. Moreover, if the initial samples are concentrated in the flat basins as claimed, doesn't that mean later there would be a period where the sampler stays around the sharp nodes? Is this because of the strongly convex assumption for Theorem 1? If this assumption is violated, to what extent does the sampler still converge to the true posterior distribution? Should I take the theoretical results as establishing the local convergence rates once we are near a mode?

It would be helpful if the authors can clarify the above issue. I am also curious to see a simple new experiment where we run EMCMC on a distribution like the one shown in Figure 2 for a really long time to see whether EMCMC would come back to the sharp mode and stay there for a while.

**Questions:**

In Table 1(a), while EMCMC performs better, it seems entropy-SGLD is almost consistently worse than SGLD. Why is that? For comparison Entropy-SGD seems to consistently outperform SGD.

---

> ### Author Response · Authors · 2023-11-19
>
> Thanks for your supportive and thoughtful comments. Please find our responses to the questions below.
>
> Q1: When we talk about sampling, we usually aim to sample from the true underlying distribution, including both the flat basins and the sharp modes.
>
> A1: We have discussed this in the introduction. Achieving an accurate estimation of the full posterior, including both flat and sharp modes, is extremely difficult in Bayesian deep learning due to its high-dimensional and highly multi-modal nature [1,2]. Under a practical budget, we cannot obtain enough samples to capture the entire posterior. We can only capture a limited number of modes (typically only 4-6 modes [1]). Therefore, we need to prioritize capturing flat basins since those samples provide good generalization and contribute significantly to Bayesian marginalization.
>
> In the case of having enough budgets, $\theta$ in our method will still converge to the original posterior, including both flat and sharp modes. We will add more explanation on this point in the final version.
>
> Q2: Theorem 1 means that $\theta$ would have to eventually visit the sharp modes.
>
> A2: It is true that the marginal distribution of $\theta$ is the original posterior, which includes both flat and sharp modes. Under a limited budget, our method prioritizes capturing flat basins because of the auxiliary variable $\theta_a$, while under infinite resources, our method ensures that $\theta$ still converges to the original posterior. This is an ideal behavior to have in Bayesian deep learning considering its highly multi-modal nature, as explained in A1.
>
> Q3: If the initial samples are concentrated in the flat basins as claimed, will there be a period where the sampler stays around the sharp modes?
>
> A3: Yes, given sufficient budgets, there will be such a period of staying at the sharp mode, since the stationary distribution of $\theta$ is the original posterior. Please check A6 for empirical results.
>
> Q4: If the strongly convex assumption is violated, to what extent does the sampler still converge to the true posterior distribution?
>
> A4: $\theta$ in EMCMC always converges to the true posterior in either convex or non-convex settings. As shown in Lemma 1, the stationary distributions of $\theta$ and $\theta_a$ in EMCMC are always guaranteed to be the original posterior and flat posterior, respectively. The strongly convex assumption is only used for analyzing convergence bounds in section 5.
>
> Q5: Should I take the theoretical results as establishing the local convergence rates once we are near a mode?
>
> A5: No, we do not have the assumption that the sampler is near a mode. The assumptions for our theorems are Assumptions 1 and 2 stated in Section 5.
>
> Q6: I am also curious to see a simple new experiment where we run EMCMC on a distribution like the one shown in Fig. 2 for a really long time to see whether EMCMC would come back to the sharp mode.
>
> A6: You can check the following links for this new experiment. We run EMCMC for a long time (20000 iterations) to collect enough samples. From the figures, we can see that, after a long time, $\theta$ will come back to the sharp mode and visit both modes (original target distribution) and $\theta_a$ will only visit the flat mode (flattened target distribution). These results align with our theoretical analysis that the stationary distribution of $\theta$ and $\theta_a$ in EMCMC are guaranteed to be the original target distribution and flat target distribution, respectively.
>
> $\theta$: https://anonymous.4open.science/r/EMCMC-FF35/sync_sampler_extended_run.pdf
>
> $\theta_a$: https://anonymous.4open.science/r/EMCMC-FF35/sync_anchor_extended_run.pdf
>
> Q7: In Table 1(a), it seems Entropy-SGLD is almost consistently worse than SGLD.
>
> A7: Entropy-SGLD is consistently worse than SGLD because of its slow convergence, which aligns with our Theorem 3. For a fair comparison, we run all methods for 200 epochs. Under this budget, Entropy-SGLD cannot converge well while our method converges faster and achieves superior performance. This demonstrates the efficiency of our method.
>
> [1] Cyclical Stochastic Gradient MCMC for Bayesian Deep Learning. ICLR 2020.
>
> [2] What Are Bayesian Neural Network Posteriors Really Like? ICML 2021.

---

> > ### Comment · Reviewer_JBLt · 2023-11-22
> > **Thanks for the response**
> >
> > I thank the authors for the detailed response. The answers addressed my concerns. I would be happy to recommend accepting the paper.

---

> > > ### Author Response · Authors · 2023-11-22
> > >
> > > Thank you for your reply! We are delighted to know we have addressed your concerns. Thank you for your helpful comments and constructive feedback again. We would appreciate it if you could consider raising the score to reflect this. Thank you!

---

### Official Review · Reviewer_nUGC · 2023-10-31

**Soundness:** 3 good
**Presentation:** 4 excellent
**Contribution:** 3 good
**Rating:** 6
**Confidence:** 4

**Summary:**

The paper describes an "augmentation" technique for Bayesian neural networks.
There is ample experimental evidence and some theoretical support that neural networks
(pointwise estimations) obtained as local minimizers of the training loss which belong to flat
basins generalize better than minimizers in sharp minima.
Therefore, in an approximate Bayesian setting where full posterior exploration is prohibitive,
they introduce an auxiliary replica, so that the two replicas measure has as marginals the original
posterior and the local entropy weight (from Baldassi et al. PRL '15).
The argument is that while the marginal distribution is still the same, the MCMC dynamics
of the replicated system allows early exploration of the wider modes of the posterior, thus obtaining
better generalization performance.

**Strengths:**

- The paper is well organized, well written, and presents some advances in the field.

- The 2 replicas framework, although very similar (actually a specific instance with y=1 replicas) of the Robust Ensemble (RE)
  introduced in Baldassi et al. PNAS '16, has the advantage over the generic RE of preserving an unbiased marginal measure,
  while in the RE with y>1 replicas the resulting marginals are tilted.

- There is good experimental coverage showing consistent (although small) improvements over competing techniques.

- They provide a favorable convergence bound for log-concave target distribution.

**Weaknesses:**

- There is a lack of novelty with respect to Baldassi et al. PNAS '16, only partly justified by the focus on the Bayesian setting.

- All experiments seem to be performed using a temperature of T=1e-4, instead of the T=1 of the purely Bayesian setting. This
  makes the Entropy-MCM framework even more similar to the optimization setting of Baldassi et al. PNAS '16 and Pittorino et al ICLR '21.
  Since table 4 shows only minimal performance decrease when setting the temperature to 1, I suggest to present all results
  with T=1 only. As an alternative, they could follow the protocol of Zhang et al. ICLR 2020b setting T=0 in the burn-in phase and T=1 in
  the sample collection phase.

- There is some hyper-parameter tuning, e.g. eta and T, carried on the test set instead of a validation set. This is not great practice.

**Questions:**

- Can the author comment on the possibility of enhancing the effect of attraction toward flat minima.
  E.g. in the framework of Baldassi et al. PNAS '16 this would mean increasing the number of replicas used.
  Can more replicas be used by also having one of the marginal equal to the origin measure?

- In some of the experiments (or maybe all) experiments the author collect samples of both theta and theta_a.
  Since the spirit of the paper is to perform Bayesian sampling, shouldn't the collect only theta samples?

- Maybe mores remands to the appendices, e.g. to appendix B for proofs, are needed in the main text.

- Fig. 6 shows a quite irregular dependence on eta of the test error, in particular the presence of very sharp peaks.
  Do the authors have any intuition of why there is this peak?

- Related to the previous question, did the author consider some adaptive tuning scheme for eta, such as the one proposed in the
 "Focusing" section of Pittorino et al ICLR '21?

- A correction for the "Related Work" section: The concept of Local Entropy has been introduced in https://journals.aps.org/prl/abstract/10.1103/PhysRevLett.115.128101.
  Entropy-SGD (Chaudhari et al., 2019) is an algorithmic implementation. (Baldassi et al., 2016) use the Local Entropy framework to derive "replicated" algorithmic approaches.
  Both Entropy-SGD and Replicated-SGD algorithms have been further investigated in https://openreview.net/forum?id=xjXg0bnoDmS

- typo in eq. 16, there should be a +

- I wonder if after the burn-in the samples are collected at each iteration.

---

> ### Author Response · Authors · 2023-11-19
>
> We appreciate your supportive and thoughtful review. Please find our responses to the questions below.
>
> Q1: There is a lack of novelty with respect to Baldassi et al. PNAS '16, only partly justified by the focus on the Bayesian setting.
>
> A1: We want to emphasize that our method, EMCMC, substantially differs from Baldassi et al.'s approach and is not a special case of the Robust Ensemble (RE) with y=1 replica. There are several key differences between EMCMC and RE in Baldassi et al.: 1) In RE, all model replicas play identical roles and possess the same stationary distribution.  In EMCMC, $\theta$ and $\theta_a$ have different stationary distributions. 2) RE approximates the integral in local entropy by a finite set of $y$ replicas of the model, with exactness achieved only in the limit of  $y\rightarrow\infty$. In contrast, EMCMC does not make any approximation by using a joint distribution in a doubled parameter space. The stationary distributions of $\theta$ and $\theta_a$ in EMCMC are guaranteed to be the original target distribution and flat target distribution respectively. 3) RE is an optimization method, aiming to find a flat optimum as the point estimation. In contrast, EMCMC is a sampling method, aiming to sample from the flat basins within the posterior.
>
> In summary, EMCMC and RE are developed from distinct ideas and they use local entropy in significantly different ways. To the best of our knowledge, the proposed auxiliary guiding variable and the joint distribution's form are novel and non-trivial solutions to flatness-aware learning. We will include these comparisons in the final version.
>
> Q2: Since Table 4 shows only minimal performance decrease when $T=1$, I suggest presenting all results with $T=1$ only.
>
> A2: In Bayesian deep learning, tempering has been widely used and is a standard technique to improve performance [1,2,4].  The performance of Bayesian neural networks typically decreases when using $T=1$, especially on large datasets. The mentioned Zhang et al. ICLR 2020b also used tempering (see Appendix J) in the sample collection phase.
>
> Moreover, EMCMC consistently outperforms previous Bayesian methods even under $T=1$. For example, on CIFAR10 with $T=1$, SGLD achieves a test accuracy of 92.40%, while EMCMC achieves 95.3%. We will add the results of $T=1$ in the final version and the above explanation on the use of tempering.
>
> Q3: There is some hyper-parameter tuning, e.g. $\eta$ and $T$, carried on the test set instead of a validation set.
>
> A3: We want to clarify that the chosen values for $\eta$ and $T$ are tuned via cross-validation, following the prior work [3]. The results presented in Table 4 and Fig. 6 serve the purpose of ablation studies, showing the influence of these hyper-parameters on performance (i.e. test accuracy). We will clarify this in the final version.
>
> Q4: Can the author comment on the possibility of enhancing the effect of attraction toward flat minima? Can more replicas be used by also having one of the marginal equal to the origin measure?
>
> A4: As discussed in A1, EMCMC differs from the methods that approximate the integral in local entropy using a finite set of replicas. EMCMC avoids the integral approximation by using a joint distribution. The stationary distributions of $\theta$ and $\theta_a$ in EMCMC are guaranteed to be the original target distribution and flat target distribution respectively. Therefore, EMCMC does not need to use more “replicas” for a more precise approximation of the integral.
>
> Q5: In some of the experiments the authors collect samples of both $\theta$ and $\theta_a$.
>
> A5: We have discussed it in Section 4.2 and Appendix C.3. Specifically, we collect both $\theta$ and $\theta_a$ in order to obtain more high-quality and diverse samples in a given time budget. The distribution of $\theta$ is the Bayesian posterior only given infinite time and computation budgets (since MCMC converges _asymptotically_ to the target distribution). Therefore, under a practical budget, we mix $\theta$ and $\theta_a$ in the sample set since both usually are from low-energy and flat regions.
>
> As shown in Appendix C.3, collecting both $\theta$ and $\theta_a$ achieves the best performance compared to using $\theta$ or $\theta_a$ only. We will add the above clarification in the final version.
>
> Q6: Maybe more remands to the appendices.
>
> A6: Thanks for the suggestion! We will add more remands in the final version.
>
> Q7: Fig. 6 shows a very sharp peak on the test error in terms of $\eta$.
>
> A7: Thanks for pointing this out! This is a typo, since it is irregular for test accuracy to reach 98% on CIFAR10. We put the corrected plot in the following link, and will update Fig. 6(a) in the final version.
>
> https://anonymous.4open.science/r/EMCMC-FF35/eta_CIFAR10_correct.pdf
>
> Please note that we choose the log-scale in Fig. (6) because we primarily want to determine the appropriate scale for $\eta$.

---

> ### Author Response · Authors · 2023-11-19
>
> Q8: Did the author consider some adaptive tuning scheme for $\eta$?
>
> A8: We did not consider adaptive tuning techniques for $\eta$ in our method. Our experiments have demonstrated that a constant value of $\eta$ already yields improvements over the baseline methods. We agree that using an adaptive tuning scheme for $\eta$ is a very interesting direction and leave it for future work.
>
> Q9: A correction for the "Related Work" section.
>
> A9: Thanks for your helpful suggestions! We will correct the statements and include the suggested related work in the final version.
>
> Q10: Typo in Eq. 16.
>
> A10: Thanks for pointing this out! We will correct it in the final version.
>
> Q11: I wonder if after burn-in the samples are collected at each iteration.
>
> A11: We collect samples at the end of each epoch after burn-in, similar to previous works [1]. This choice is mainly due to memory constraints for deep neural networks.
>
> [1] Cyclical Stochastic Gradient MCMC for Bayesian Deep Learning. ICLR 2020.
>
> [2] How good is the bayes posterior in deep neural networks really? ICML 2020.
>
> [3] What Are Bayesian Neural Network Posteriors Really Like? ICML 2021.
>
> [4] Preconditioned Stochastic Gradient Langevin Dynamics for Deep Neural Networks. AAAI 2016.

---

> ### Comment · Reviewer_nUGC · 2023-11-20
> **Proposed method is Robust Ensemble with y=1**
>
> > A1: We want to emphasize that our method, EMCMC, substantially differs from Baldassi et al.'s approach and is not a special case of the Robust Ensemble (RE) with y=1 replica. There are several key differences between EMCMC and RE in Baldassi et al.: 1) In RE, all model replicas play identical roles and possess the same stationary distribution. In EMCMC,
>  and
>  have different stationary distributions.
>
> I have to insist on the fact that the proposed method is exactly the RE with y=1. This is immediately seen by considering Eq. 3 of
> https://www.pnas.org/doi/full/10.1073/pnas.1608103113 and setting y=1. One gets a system of two replicas, the "central one" ($\sigma^*$ in the PNAS),  and the peripheral one $\sigma^1$, playing exactly the same role as $\theta_a$ and $\theta$ in this work.
>
> > 2) RE approximates the integral in local entropy by a finite set of
>  replicas of the model, with exactness achieved only in the limit of
> . In contrast, EMCMC does not make any approximation by using a joint distribution in a doubled parameter space. The stationary distributions of
>  and
>  in EMCMC are guaranteed to be the original target distribution and flat target distribution respectively.
>
> This comment indicates a complete misunderstanding by the authors of the main concepts in the RE paper. RE is not used to approximate the local entropy integral. It's an auxiliary system that generalizes the one of the current paper and in which the statical weight at the exponent is the local entropy times a factor y (y plays the role of an inverse temperature). I invite the authors to read (carefully) the paper.
>
> > 3) RE is an optimization method, aiming to find a flat optimum as the point estimation. In contrast, EMCMC is a sampling method, aiming to sample from the flat basins within the posterior.
>
> I agree on this point, and I already appreciated the different (Bayesian) perspective in the original comment.
>
>
> > A4: As discussed in A1, EMCMC differs from the methods that approximate the integral in local entropy using a finite set of replicas. EMCMC avoids the integral approximation by using a joint distribution. The stationary distributions of
>  and
>  in EMCMC are guaranteed to be the original target distribution and flat target distribution respectively. Therefore, EMCMC does not need to use more “replicas” for a more precise approximation of the integral.
>
> Given the comments above, the authors could reconsider this question. My point is that one can give more statistical weight to the local entropy using an inverse temperature, $\exp(y \mathcal{L}_{LE})$, and then applying the replica trick as in the RE or in this paper. But likely if $y\neq 1$ the fact of having one replica with the the origin posterior as marginal is lost.

---

> ### Author Response · Authors · 2023-11-20
> **Response to follow-up questions**
>
> Q1: Proposed method is Robust Ensemble with y=1
>
> A1: Thank you very much for your reply and insightful comments! However, we believe there are key differences between EMCMC and Robust Ensemble with y=1:
>
> - In Baldassi et al. PNAS '16, the definition of Robust Ensemble, as outlined in Eq. 4, traces out the reference ($\theta_a$ in our paper) and only has $y$ identical replicas in the system, resembling ensemble methods. This can be further verified by Pittorino et al. ICLR 2021 (https://openreview.net/forum?id=xjXg0bnoDmS). As shown in Algorithm 2 in Pittorino et al. ICLR 2021, Robust Ensemble with y=1 will essentially be the same as standard SGD.
>
>    In contrast, EMCMC has two variables performing different roles, unlike ensemble methods. $\theta_a$ cannot be traced out and plays a crucial role in leading to flat basins during training.
>
>   Furthermore, Robust Ensemble typically requires at least three models (i.e. $y\ge 3$) while EMCMC only needs two models. Thus, EMCMC significantly reduces computational costs, which is especially beneficial when using large deep neural networks.
>
> - We acknowledge that Eq. 3 with y=1 in Baldassi et al. PNAS '16 is equivalent to the proposed joint distribution. We will clarify this point in the final version. However, Baldassi et al. PNAS '16 did not consider the marginal distribution of replicas. In contrast, our work highlights that the marginal of the replica is the original target distribution. This distinction arises because our joint distribution is derived by coupling the original and flattened distributions, a different motivation from Baldassi et al. PNAS '16. Moreover, to the best of our knowledge, Eq. 3 has not been practically applied (as seen in Baldassi et al. PNAS '16 and Pittorino et al. ICLR 2021), primarily serving as an intermediate step in deriving Robust Ensemble (which uses $y$ identical replicas).
>
>
>
> Your valuable comments greatly help clarify the relationship between EMCMC and existing literature. We will add the above discussion in the final version. We believe this work introduces novel perspectives on local entropy and an efficient algorithm accompanied by theoretical and empirical results. We would appreciate it if these contributions and the potential impact of this work could be considered in your final assessment.
>
>
> Q2: Can the author comment on the possibility of enhancing the effect of attraction toward flat minima. Can more replicas be used by also having one of the marginal equal to the origin measure?
>
> A2: Thank you for the further explanation! We think an appropriate tempering for the flattened posterior can potentially enhance the effect of attraction toward flat basins. This is similar to the use of tempering in Bayesian learning [1,2] to improve inference results. However, unlike Robust Ensemble, a small temperature (i.e. a large $y$) may not always be helpful, since it may potentially result in the loss of uncertainty information in the posterior. Furthermore, keeping the original target distribution as one replica's marginal distribution may be a non-trivial question. We think this is an interesting direction and we are happy to explore it further in future work.
>
> Please let us know if you have any further questions, and we will be happy to respond.
>
> [1] Inconsistency of Bayesian inference for misspecified linear models, and a proposal for repairing it. Bayesian Analysis 2017.
>
> [2] Parallel tempering: Theory, applications, and new perspectives. Physical Chemistry Chemical Physics 2005.

---

> ### Author Response · Authors · 2023-11-22
>
> We are grateful for your helpful comments and constructive feedback on our paper. Please kindly let us know if you have any remaining questions or concerns so that we can address them before the deadline.
>
> If you feel that your original concerns have been addressed, we would appreciate it if you could consider raising the score to reflect this. Thank you!

---

### Official Review · Reviewer_QrYa · 2023-11-01

**Soundness:** 3 good
**Presentation:** 4 excellent
**Contribution:** 3 good
**Rating:** 8
**Confidence:** 4

**Summary:**

This paper studies the flatness aware optimization using MCMC with an entropy-adjusted loss function. The overall goal is to adapt SGD with information about the local flatness of the optimization landscape in order to find local minima in flatter regions of the loss landscape, which are widely believed to have better generalization properties compared to sharp local minima. The key technical insight is that the computational difficulties encountered applying local entropy optimization in previous works can be avoided by performing joint MCMC on the energy term inside the local entropy integral, whose marginal distributions correspond to the original Bayesian posterior distribution and the local entropy posterior distribution respectively. Joint sampling allows the local entropy weight path to facilitate movement of the original weight path. A theoretical analysis shows the proposed method has more favorable convergence rates than prior local entropy optimization methods in the strongly convex setting. Experiments show that the proposed method finds local minimum which exhibit greater flatness according to the eigenspectrum and interpolation/extrapolation experiments, and that the proposed method boosts validation accuracy compared to SGD and related entropy-based optimization methods on CIFAR-10, CIFAR-100, and ImageNet.

**Strengths:**

* The paper is easy to understand and well written.
* The method resolves a key computational limitation in the local entropy approach from Chaudhari et. al 2019 and allows local entropy optimization with no inner loop, and essentially the same computational cost as SGD. This could make local entropy optimization much more appealing to practitioners than current methods.
* Synthetic dataset experiments and measurement of flatness metrics corroborate the claims of the method's ability to focus its trajectory on flat minima.
* Experiments on classifier training show improved performance relative to SGD and existing local entropy optimization methods.

**Weaknesses:**

* The theoretical results focus on a very restricted case of strong convexity. Although analysis of this situation provides interesting context for the relative abilities of the proposed method and existing methods, nothing can be firmly concluded in realistic settings.
* The SGD baselines for the classification experiments are somewhat weak. It would be interesting to see if the proposed method can push the performance of models with state of the art scores, or at least much closer to state of the art. Maybe fine-tuning rather than full training could help alleviate costs.
* The degree of novelty is not especially high, as the method is a straightforward change to the approach from Chaudhary et. al 2019 and the experimental settings are fairly commonplace. On the other hand, the simplicity of the proposed method is part of its appeal.

**Questions:**

Many works find that flat minima lead to better generalization, but some works such as Dinh et al. 2017 (cited in the paper) claim that flatness is not necessarily for good generalization. Can the authors elaborate on the reasons for believing that flatness leads to better generalization and discuss whether generalization can be (or cannot be) achieved without flatness?

---

> ### Author Response · Authors · 2023-11-19
>
> Thanks for your supportive and valuable comments. Please find our responses to the questions below.
>
> Q1: The theoretical results focus on a very restricted case of strong convexity.
>
> A1: Strong convexity is a common assumption in analyzing the convergence of stochastic gradient MCMC, e.g. in [5,6,7]. Our theoretical results follow this line of works and provide convergence guarantees for the proposed method. Under the strongly convex setting, there are realistic tasks (e.g., logistic regression in Section 6.2) and we can firmly conclude that our method converges faster than baselines on these tasks. We agree that the convergence analysis under non-convex settings is a very interesting research question and we leave it to future work.
>
> Q2: The SGD baselines for the classification experiments are somewhat weak.
>
> A2: The SGD baselines are aligned with previous works in Bayesian deep learning such as [1,8,9]. We used the same model architecture and training procedure for all methods and did not adopt additional engineering tricks to get better performance. This ensures that our SGD baselines serve as a fair benchmark for comparison against our proposed method.
>
> Q3: It would be interesting to see if the proposed method can push the performance of models with state of the art scores.
>
> A3: Our experimental results demonstrate the superior performance of our method compared to prior state-of-the-art (SOTA) sampling methods in Bayesian deep learning, such as [1]. Achieving SOTA results in general deep learning often involves large models and numerous engineering tricks. To ensure a clear and fair comparison, we conducted experiments with standard models and benchmarks, since achieving SOTA is not the main focus of this work. However, we agree that testing with SOTA models and pushing the SOTA scores are interesting questions. We will apply our method to large pre-trained models and report the results in the final version.
>
> Q4: The method is a straightforward change to the approach from Chaudhary et. al 2019.
>
> A4: We would like to emphasize that our method is not a straightforward change from Chaudhary et. al 2019. In fact,  it is highly non-trivial to develop an efficient flatness-aware method, as biasing toward the flat basins often introduces substantial computational overhead [10, 4].
>
> A key issue with Chaudhary et al. 2019 approach is its usage of nested Markov chains with Monte Carlo approximation, which significantly reduces convergence speed and estimation accuracy. In contrast, our method, by using an auxiliary variable, guarantees to converge to flat basins with fast speed and minimal overhead. To the best of our knowledge, the proposed auxiliary guiding variable and the joint distribution's form are novel and non-trivial solutions to flatness-aware learning. We believe that the simplicity of our method should not diminish its novelty. We appreciate your acknowledgment of the benefits from the simplicity of our approach.
>
> Q5: The experimental settings are fairly commonplace
>
> A5: We believe that the experiments are standard and comprehensive enough to demonstrate the effectiveness and efficiency of EMCMC. Our experimental settings follow previous works [1,10,11,12,13]. Specifically, we show that EMCMC can find flat modes in Sections 6.1&6.3, verify the fast convergence of EMCMC in Section 6.2, demonstrate the classification performance in Section 6.4, and the uncertainty estimation performance in Section 6.5. We agree that applying our method to more ML tasks, beyond standard models and applications, is an interesting question and we leave it for future work.

---

> ### Author Response · Authors · 2023-11-19
>
> Q6: Can the authors elaborate on the reasons for believing that flatness leads to better generalization and discuss whether generalization can be (or cannot be) achieved without flatness?
>
> A6: In the second paragraph of the introduction, we mentioned the reason for flatness leads to better generalization from a Bayesian perspective. Specifically, to make prediction for testing data point $x^*$ in Bayesian learning, we need to obtain the following predictive distribution:
> $$
> p(y^*|x^*,D)=\int p(y^*|x^*,\theta)p(\theta|D)d\theta.
> $$
> Samples of $\theta$ from the flat basin within the posterior occupy a substantial volume and significantly contribute to this integral. Therefore, obtaining $\theta$ from the flat basin can lead to a more precise estimation of the posterior and the predictive distribution. This will lead to better generalization according to PAC-Bayes bounds [2, 3].
>
> However, comprehensively justifying the relationship between flatness and generalization still remains an open question and beyond the scope of this work.
>
> [1] Cyclical Stochastic Gradient MCMC for Bayesian Deep Learning. ICLR 2020.
>
> [2] Entropy-SGD optimizes the prior of a PAC-Bayes bound: Generalization properties of Entropy-SGD and data-dependent priors. ICML 2018.
>
> [3] PAC-Bayesian stochastic model selection. Machine Learning 2003.
>
> [4] Sharpness-aware minimization for efficiently improving generalization. ICLR 2020.
>
> [5] User-friendly Guarantees for the Langevin Monte Carlo with Inaccurate Gradient. Stochastic Processes and their Applications, 2019.
>
> [6] On the Theory of Variance Reduction for Stochastic Gradient Monte Carlo. ICML 2018.
>
> [7] On the Convergence of Hamiltonian Monte Carlo with Stochastic Gradients. ICML 2021.
>
> [8] Bayesian image classification with deep convolutional Gaussian processes. AISTATS, 2020.
>
> [9] What Are Bayesian Neural Network Posteriors Really Like? ICML 2021.
>
> [10] Entropy-SGD: Biasing Gradient Descent into Wide Valleys. ICLR 2017.
>
> [11] Averaging Weights Leads to Wider Optima and Better Generalization. UAI 2018.
>
> [12] Asymptotically Optimal Exact Minibatch Metropolis-Hastings. NeurIPS 2020.
>
> [13] Addressing Failure Prediction by Learning Model Confidence. NeurIPS 2019.

---

> > ### Comment · Reviewer_QrYa · 2023-11-22
> > **Thanks for the discussion. I will keep my score.**
> >
> > Thanks to the authors for their authoritative discussion with myself and other reviewers. I will keep my score and I am glad to recommend this paper.

---

> > > ### Author Response · Authors · 2023-11-22
> > >
> > > Thank you for your response. We appreciate your insightful comments and acknowledgment of our paper!

---

### Official Review · Reviewer_CmLw · 2023-11-04

**Soundness:** 3 good
**Presentation:** 3 good
**Contribution:** 3 good
**Rating:** 6
**Confidence:** 4

**Summary:**

The authors define the augmented model where they aim to perform the inference over variable $\theta_a$ in addition to model parameters.

The authors show the marginal posterior of the augmented model over model parameters give the "right" posterior distribution.

The authors provide several theoretical statements justifying the algorithm and the empirical evaluation of the algorithm.

**Strengths:**

I think the idea of the augmented model is interesting and the paper is a nice read.

These considerations are novel to the best of my knowledge.

The manuscript is mostly very clear.

The proposed method achieves good empirical performance.

**Weaknesses:**

I think the authors should clearly specify their model: prior distributions and likelihood and only after that move to the inference part to improve the clarity of the paper.

I understand that when priors are uniform, the RHS of Eqn 4 effectively defines the likelihood of the augmented model the authors want to consider.

The notation using $f(\theta)$ is confusing (e.g. because of no dependence on data) and should be replaced by substituting the definition above the Eq 3.

No line numbers in the manuscript.

Is the variance $\eta$ a free variable? Why does $\eta$  in Eqn 21 disappear during integration?

It is not clear how data augmentation influences the empirical performance of the method.

It would be nice if the authors provided some arguments why the introduced concepts are mathematically well-defined, e.g. (4) integrates to a finite quantity (seems straightforward but the finiteness of all quantities should be ensured).

Why do authors consider the assumption 1? I.e. it's fair to say that authors what to perform the inference in the augmented model, specify the prior/likelihood and there's no need to introduce the assumption 1 (make it a remark).

It's difficult to judge the practical utility of theoretical statements.

If my questions are addressed I'm willing to increase my score.

**Questions:**

See the weaknesses section.

---

> ### Author Response · Authors · 2023-11-19
>
> Thanks for your supportive and thoughtful comments. Please find our responses to the questions below.
>
> Q1: The authors should mention prior distributions and likelihood and then move to the inference part.
>
> A1: Thanks for the suggestion. In Section 3 Preliminaries (the SGMCMC paragraph), we explicitly discussed the probabilistic model used in the paper. Specifically, we consider a neural network with parameters $\theta$. The prior distribution is denoted as $p(\theta)$, and the energy function is denoted as $U(\theta)$. We will mention explicitly that the likelihood is denoted as $p(D|\theta)$. Similar to prior works in Bayesian deep learning (e.g. [6, 7]), we do not introduce additional assumptions regarding the prior or likelihood. We will include the above clarification in the final version to improve the clarity of the paper.
>
> Q2: I understand that when priors are uniform, the RHS of Eqn 4 effectively defines the likelihood of the augmented model the authors want to consider.
>
> A2: As mentioned in Section 4.1, the joint distribution in Eq.4 is obtained by coupling two marginal distributions, the original posterior and the flattened posterior. It is important to note that the joint distribution is _not_ obtained by Bayes’ rule (which is obtained through the prior and the likelihood). We will further clarify this in the final version.
>
> Q3: The notation of $f(\theta)$ is not dependent on data.
>
> A3: Existing works in both flatness-aware optimization (e.g., [1, 3]) and Bayesian deep learning (e.g., [6, 7]) typically use the notation $f(\theta)$ to denote the loss function and the energy function. To align with the existing literature, we use the same notation. The omission of explicit data dependency in this notation is because data is considered fixed. The energy (or loss) function mainly operates as a function of $\theta$ during training. To enhance clarity, we will add a sentence to make it clear that $f(\theta)$ depends on training data.
>
> Q4: No line numbers in the manuscript.
>
> A4: To the best of our knowledge, the ICLR submission template does not include line numbers.
>
> Q5: It would be nice if the authors provided some arguments why the introduced concepts are mathematically well-defined.
>
> A5: The introduced joint distribution in Eq. 4 is mathematically well-defined. To see so, suppose the original posterior is $p(\theta|D) = \frac{1}{Z}\exp(-f(\theta))$, then the normalizing constant for Eq. 4 is
> $$
> \int\int\exp(-f(\theta)-\frac{1}{2\eta}\\|\theta-\theta_a\\|^2)d\theta_a d\theta=(2\pi\eta)^{\frac{d}{2}}\int\exp(-f(\theta))d\theta=(2\pi\eta)^{\frac{d}{2}}Z.
> $$
> This shows that the normalizing constant is finite and thus Eq. 4 is mathematically well-defined. We will add the above explanation in the final version.
>
> To the best of our knowledge, a well-defined distribution does not require the finiteness of all quantities (e.g. Cauchy distribution). Therefore, we think ensuring the finiteness of all quantities is not necessary. Should there be any misunderstanding, please let us know and we would be happy to offer further clarification.
>
> Q6: Is the variance $\eta$ a free variable? Why does $\eta$ in Eq. 21 disappear during integration?
>
> A6: $\eta$ is a hyperparameter in our algorithm. In Eq. 21, $\eta$ disappears since the marginal distribution of $\theta$ does not depend on $\eta$. To illustrate this, suppose the original posterior is $p(\theta|D) = \frac{1}{Z}\exp(-f(\theta))$ and then the joint distribution can be represented as $p(\theta,\theta_a|D) = (2\pi\eta)^{-\frac{d}{2}}Z^{-1}\exp(-f(\theta)-\frac{1}{2\eta}\\|\theta-\theta_a\\|^2)$. Eq. 21 becomes
> $$
> \begin{align}
>     \int p(\theta,\theta_a|D)d\theta_a
>     &=(2\pi\eta)^{-\frac{d}{2}}Z^{-1}\int\exp(-f(\theta)-\frac{1}{2\eta}\\|\theta-\theta_a\\|^2)d\theta_a \\\\
>     &=Z^{-1}\exp(-f(\theta))(2\pi\eta)^{-\frac{d}{2}}\int\exp(-\frac{1}{2\eta}\\|\theta-\theta_a\\|^2)d\theta_a \\\\
>     &=Z^{-1}\exp(-f(\theta)) \\\\
>     &=p(\theta|D)
> \end{align}
> $$
> The above derivation clearly shows that the marginal distribution of $\theta$ is equal to the original posterior distribution and does not depend on $\eta$. In our final version, we will include this explanation for clarity.
>
> Q7: It is not clear how data augmentation influences the empirical performance of the method.
>
> A7: We have studied the influence of data augmentation on our method in Appendix A.2. Specifically, we found that: 1) With or without data augmentation, EMCMC consistently outperforms baseline methods. 2) The improvement of EMCMC over baselines is larger in the case without data augmentation. 3) With data augmentation, all methods achieve higher accuracy. In summary, these results clearly show how data augmentation influences the empirical performance of our method EMCMC, and demonstrate that EMCMC can consistently provide improvement with or without data augmentation.

---

> ### Author Response · Authors · 2023-11-19
>
> Q8: Why do authors consider Assumption 1?
>
> A8: We would like to emphasize that Assumption 1 is used only for the theoretical analysis in Section 5. Our methodology and experiments are not restricted to this assumption. Assumption 1 is a common assumption in analyzing the convergence of stochastic gradient MCMC (e.g. [2,4,5]). Following existing literature, we provide convergence bounds of our method in the strongly convex setting in the paper. We agree that the convergence analysis of our method under non-convex settings is a very interesting question and we leave it to future work.
>
> Q9: It's difficult to judge the practical utility of theoretical statements.
>
> A9: The practical utility of our theoretical statements can be summarized into three aspects: 1) They provide rigorous guarantees on the convergence of our method and further demonstrate the superior convergence rate of our method compared to previous methods in the strongly convex setting. These results are also verified empirically in Fig.3. 2) They formally demonstrate the advantages of using the proposed joint distribution over the nested Markov chains in previous works. Since the nested chains induce a large and complicated error term $A$ as shown in our theorems. 3) They provide insights into why our method converges faster and shed light on future theoretical analysis in the non-convex setting. We hypothesize that the advantage of our method extends similarly to the non-convex setting, which is empirically supported by our experimental results.
>
> In summary, we believe our theoretical statements have sufficient practical utility. They offer theoretical guarantees for the proposed methodology, rigorously justify the advantages of the joint distribution over nested chains, and provide insights on future analysis in non-convex settings.
>
> [1] Entropy-SGD: Biasing Gradient Descent into Wide Valleys. ICLR 2017.
>
> [2] User-friendly Guarantees for the Langevin Monte Carlo with Inaccurate Gradient. Stochastic Processes and their Applications, 2019.
>
> [3] Entropy-SGD optimizes the prior of a PAC-Bayes bound: Generalization properties of Entropy-SGD and data-dependent priors. ICML 2018.
>
> [4] On the Theory of Variance Reduction for Stochastic Gradient Monte Carlo. ICML 2018.
>
> [5] On the Convergence of Hamiltonian Monte Carlo with Stochastic Gradients. ICML 2021.
>
> [6] Stochastic Gradient Hamiltonian Monte Carlo. ICML 2014.
>
> [7] Cyclical Stochastic Gradient MCMC for Bayesian Deep Learning. ICLR 2020.

---

> > ### Author Response · Authors · 2023-11-22
> >
> > We are grateful for your helpful comments and constructive feedback on our paper. Please kindly let us know if you have any remaining questions or concerns so that we can address them before the deadline.
> >
> > If you feel that your original concerns have been addressed, we would appreciate it if you could consider raising the score to reflect this. Thank you!

---

### Author Response · Authors · 2023-11-21
**Summary of Responses**

We thank all reviewers for their constructive reviews and appreciate the unanimous acknowledgment of the significance of this work. According to reviewers’ comments, we have revised the paper and highlighted the updates in the paper PDF in blue. We will move the updates in Appendix to the main body given one extra page in the final version.

*  Based on suggestions by Reviewers CmLw and Prwy: we added clarification on the definitions, the notation, and the derivation in Eq. 21.
* Based on suggestions by Reviewers CmLw, QrYa, JBLt, and Prwy: we added explanations on the significance of our theorems, the use of Assumption 1, and the convergence behavior of EMCMC. We also added additional visualizations to support our explanation.
* Based on suggestions by Reviewers QrYa and nUGC: we added additional discussion of EMCMC with Chaudhary et. al 2019 and Baldassi et al. PNAS '16.
* Based on suggestions by Reviewers nUGC: we added more details on the experiment settings, the use of tempering, and the way we tune hyperparameters.

We believe that this work provides significant and timely contributions to both Bayesian deep learning and deep learning generalization, including novel methodology, a highly practical algorithm, theoretical analysis on convergence rates, and comprehensive experimental results. We hope reviewers can consider our response in their final evaluation.

---

### Meta-Review · Area_Chair_Gkr7 · 2023-12-06

**Metareview:**

This paper proposes a new MCMC sampling method for BNN posteriors, which is biased towards flatter minima of the loss landscape, arguing that those tend to generalize better than the sharper ones. After an active discussion between authors and reviewers, the overall assessment is rather positive, with four reviewers leaning towards acceptance and only one (weakly) learning towards rejection. While the reviewers praised the novelty, empirical performance, simplicity of the idea, and convergence results, they were critical of the clarity of the theory, the restrictive setting of strong convexity, and the open questions regarding convergence to the true vs. smoothened posterior. However, most (if not all) of these issues have been addressed in the extensive rebuttal. It therefore seems warranted to accept the paper, with the understanding that the authors will continue their efforts to fully address the reviewer feedback in the camera-ready version.

**Justification For Why Not Higher Score:**

it is unclear that this work would be interesting for a wide enough audience to warrant a spotlight

**Justification For Why Not Lower Score:**

the authors addressed most of the reviewers' concerns during the rebuttal and most reviewers agree that it should be accepted

---

### Decision · Program_Chairs · 2024-01-16

Accept (poster)